# Modelling the impact of decidual senescence on embryo implantation in human endometrial assembloids

**Thomas M Rawlings**[1,2]**, Komal Makwana**[1,2]**, Deborah M Taylor**[1,2,3]**, Matteo A Molè**[4]**, Katherine J Fishwick**[1]**, Maria Tryfonos**[1,2]**, Joshua Odendaal**[1,5]**, Amelia Hawkes**[1,5]**, Magdalena Zernicka-Goetz**[4,6]**, Geraldine M Hartshorne**[1,2,3]**, Jan J Brosens**[1,2,5]*****, Emma S Lucas**[1,2]

[1]Division of Biomedical Sciences, Warwick Medical School, University of Warwick, Coventry, United Kingdom; [2]Centre for Early Life, Warwick Medical School, University of Warwick, Coventry, United Kingdom; [3]Centre for Reproductive Medicine, University Hospitals Coventry and Warwickshire NHS Trust, Coventry, United Kingdom; [4]Department of Physiology, Development and Neuroscience, University of Cambridge, Cambridge, United Kingdom; [5]Tommy's National Centre for Miscarriage Research, University Hospitals Coventry & Warwickshire NHS Trust, Coventry, United Kingdom; [6]Synthetic Mouse and Human Embryology Group, California Institute of Technology (Caltech), Division of Biology and Biological Engineering, Pasadena, United Kingdom

*For correspondence:
J.J.Brosens@warwick.ac.uk

**Abstract** Decidual remodelling of midluteal endometrium leads to a short implantation window after which the uterine mucosa either breaks down or is transformed into a robust matrix that accommodates the placenta throughout pregnancy. To gain insights into the underlying mechanisms, we established and characterized endometrial assembloids, consisting of gland-like organoids and primary stromal cells. Single-cell transcriptomics revealed that decidualized assembloids closely resemble midluteal endometrium, harbouring differentiated and senescent subpopulations in both glands and stroma. We show that acute senescence in glandular epithelium drives secretion of multiple canonical implantation factors, whereas in the stroma it calibrates the emergence of anti-inflammatory decidual cells and pro-inflammatory senescent decidual cells. Pharmacological inhibition of stress responses in pre-decidual cells accelerated decidualization by eliminating the emergence of senescent decidual cells. In co-culture experiments, accelerated decidualization resulted in entrapment of collapsed human blastocysts in a robust, static decidual matrix. By contrast, the presence of senescent decidual cells created a dynamic implantation environment, enabling embryo expansion and attachment, although their persistence led to gradual disintegration of assembloids. Our findings suggest that decidual senescence controls endometrial fate decisions at implantation and highlight how endometrial assembloids may accelerate the discovery of new treatments to prevent reproductive failure.

## Introduction

Upon embryo implantation, the cycling human endometrium transforms into the decidua of pregnancy to accommodate the placenta (*Gellersen and Brosens, 2014*). Transition between these physiological endometrial states requires intensive tissue remodelling, a process termed decidualization. Notwithstanding that decidualization in early pregnancy cannot be studied directly, a spectrum of prevalent reproductive disorders is attributed to perturbations in this process, including recurrent

**eLife digest** At the beginning of a human pregnancy, the embryo implants into the uterus lining, known as the endometrium. At this point, the endometrium transforms into a new tissue that helps the placenta to form. Problems in this transformation process are linked to pregnancy disorders, many of which can lead to implantation failure (the embryo fails to invade the endometrium altogether) or recurrent miscarriages (the embryo implants successfully, but the interface between the placenta and the endometrium subsequently breaks down).

Studying the implantation of human embryos directly is difficult due to ethical and technical barriers, and animals do not perfectly mimic the human process, making it challenging to determine the causes of pregnancy disorders. However, it is likely that a form of cellular arrest called senescence, in which cells stop dividing but remain metabolically active, plays a role. Indeed, excessive senescence in the cells that make up the endometrium is associated with recurrent miscarriage, while a lack of senescence is associated with implantation failure.

To study this process, Rawlings et al. developed a new laboratory model of the human endometrium by assembling two of the main cell types found in the tissue into a three-dimensional structure. When treated with hormones, these 'assembloids' successfully mimic the activity of genes in the cells of the endometrium during implantation. Rawlings et al. then exposed the assembloids to the drug dasatinib, which targets and eliminates senescent cells. This experiment showed that assembloids become very robust and static when devoid of senescent cells.

Rawlings et al. then studied the interaction between embryos and assembloids using time-lapse imaging. In the absence of dasatinib treatment, cells in the assembloid migrated towards the embryo as it expanded, a process required for implantation. However, when senescent cells were eliminated using dasatinib, this movement of cells towards the embryo stopped, and the embryo failed to expand, in a situation that mimics implantation failure.

The assembloid model of the endometrium may help scientists to study endometrial defects in the lab and test potential treatments. Further work will include other endometrial cell types in the assembloids, and could help increase the reliability of the model. However, any drug treatments identified using this model will need further research into their safety and effectiveness before they can be offered to patients.

implantation failure and recurrent pregnancy loss (*Dimitriadis et al., 2020*; *Macklon, 2017*; *Zhou et al., 2019*). By contrast, the sequence of events that renders the endometrium receptive to embryo implantation has been investigated extensively, starting with obligatory oestrogen-dependent tissue growth following menstrual repair. As a consequence of rapid proliferation of stromal fibroblasts and glandular epithelial cells (EpCs), which peaks in the upper third of the functional layer (*Ferenczy et al., 1979*), endometrial volume and thickness increases multifold prior to ovulation (*Raine-Fenning et al., 2004*; *Dallenbach-Hellweg, 1981*). After the postovulatory rise in progesterone levels, proliferation of EpCs first decreases and then ceases altogether in concert with the onset of apocrine glandular secretions, heralding the start of the midluteal window of implantation (*Dallenbach-Hellweg, 1981*). Concurrently, uterine natural killer (uNK) cells accumulate and endometrial stromal cells (EnSCs) start decidualizing in a process that can be described as 'inflammatory programming' (*Brighton et al., 2017*; *Chavan et al., 2021*; *Erkenbrack et al., 2018*; *Salker et al., 2012*). Morphological decidual cells, characterized by abundant cytoplasm and enlarged nuclei, emerge upon closure of the 4-day implantation window, meaning that the endometrium has become refractory to embryo implantation (*Gellersen and Brosens, 2014*). In pregnancy, decidual cells form a robust, tolerogenic matrix in which invading trophoblast cells cooperate with local immune cells to form a haemochorial placenta (*Aplin et al., 2020*; *Vento-Tormo et al., 2018*). In non-conception cycles, however, falling progesterone levels and influx of neutrophils lead to breakdown of the superficial endometrial layer and menstrual shedding (*Jabbour et al., 2006*).

Recently, we highlighted the importance of cellular senescence in endometrial remodelling during the midluteal implantation window (*Brighton et al., 2017*; *Lucas et al., 2020*; *Kong et al., 2021*). Senescence denotes a cellular stress response triggered by replicative exhaustion or other stressors that cause macromolecular damage (*Muñoz-Espín and Serrano, 2014*). Activation of tumour

suppressor pathways and upregulation of cyclin-dependent kinase inhibitors p16[INK4a] (encoded by *CDKN2A*) and p21[CIP1] (*CDKN1A*) lead to permanent cell cycle arrest, induction of survival genes, and production of a bioactive secretome, referred to as the senescence-associated secretory phenotype (SASP). The composition of the SASP is tissue-specific but typically includes proinflammatory and immunomodulatory cytokines, chemokines, growth factors, and extracellular matrix (ECM) proteins and proteases (*Birch and Gil, 2020*). Acute senescence, characterized by transient SASP production and rapid immune-mediated clearance of senescent cells, is widely implicated in processes involving physiological tissue remodelling, including during embryo development, placenta formation, and wound healing (*Muñoz-Espín and Serrano, 2014*; *Van, 2014*). By contrast, persisting senescent cells cause chronic inflammation or 'inflammaging' (*Birch and Gil, 2020*), a pathological state that underpins ageing and age-related disorders. We demonstrated that inflammatory reprogramming of EnSC burdened by replication stress leads to the emergence of acute senescent cells during the implantation window (*Brighton et al., 2017*; *Lucas et al., 2020*; *Kong et al., 2021*). Upon successful implantation and continuous progesterone signalling, decidual cells co-opt uNK cells to eliminate their senescent counterparts through granule exocytosis (*Brighton et al., 2017*; *Lucas et al., 2020*; *Kong et al., 2021*). Clearance of senescent decidual cells likely necessitates recruitment of bone marrow-derived decidual precursor cells, which confer tissue plasticity for rapid decidual expansion in early pregnancy (*Diniz-da-Costa et al., 2021*). Importantly, lack of clonogenic decidual precursor cells and a pro-senescent decidual response are linked to recurrent pregnancy loss (*Lucas et al., 2016*; *Lucas et al., 2020*; *Tewary et al., 2020*).

Based on these insights, we hypothesized that acute senescence is integral to successful implantation by creating conditions for anchorage of the conceptus in an otherwise tightly adherent decidual matrix. To test this hypothesis, we developed an 'assembloid' model, consisting of endometrial gland-like organoids and primary EnSC, which recapitulates the complexity in cell states and gene expression of the midluteal implantation window, improving resemblance to endometrial tissue in comparison with existing co-culture models (*Cheung et al., 2021*; *Rawlings, 2021*). We used this model to establish co-cultures with human blastocysts and demonstrate that aspects of different pathological states associated with implantation failure and miscarriage can be recapitulated in endometrial assembloids by modulating decidual senescence.

## Results
### Establishment of endometrial assembloids

Organoids consisting of gland-like structures are established by culturing endometrial EpCs seeded in Matrigel in a chemically defined medium containing growth factors and signal transduction pathway modulators (*Supplementary file 1*: Table 1; *Turco et al., 2017*; *Boretto et al., 2017*). Gland-like organoids grown in this medium, termed expansion medium, are genetically stable, easily passaged, and can be maintained in long-term cultures (*Boretto et al., 2017*; *Turco et al., 2017*). Oestradiol (E2) promotes proliferation of gland-like organoids and cooperates with NOTCH signalling to activate ciliogenesis in a subpopulation of EpC (*Haider et al., 2019*). Further, treatment with a progestin (e.g. medroxyprogesterone acetate [MPA]) and a cyclic AMP analogue (e.g. 8-bromo-cAMP) induces secretory transformation of gland-like organoids in parallel with expression of luteal-phase marker genes (*Turco et al., 2017*; *Boretto et al., 2017*).

We modified the gland-like organoid model to incorporate EnSC. To this end, midluteal endometrial biopsies (*Supplementary file 1*: Table 2) were digested and gland-like organoids established from isolated EpC (*Figure 1A*). In parallel, purified EnSC were propagated in standard monolayer cultures. At passage 2, single-cell suspensions of EnSC were combined with organoid EpC, seeded in hydrogel, and cultured in expansion medium supplemented with E2 (*Figure 1A*). The hydrogel matrix comprised 97% type I and 3% type III collagens, which are both present in midluteal endometrium (*Oefner et al., 2015*; *Aplin et al., 1988*; *Aplin and Jones, 1989*; *Iwahashi et al., 1996*), and has a predicted in-use elastic modulus (Pa) of comparable magnitude to non-pregnant endometrium (*Abbas et al., 2019*; *Bagley, 2019*). As shown in *Figure 1B*, gland formation was unperturbed by the presence of EnSC and assembloids resembled the architecture of native endometrium more closely than organoids. Further, decidualization of assembloids with 8-bromo-cAMP and MPA for 4 days (*Figure 1C*) resulted in robust secretion of decidual prolactin (PRL) and C-X-C motif chemokine ligand

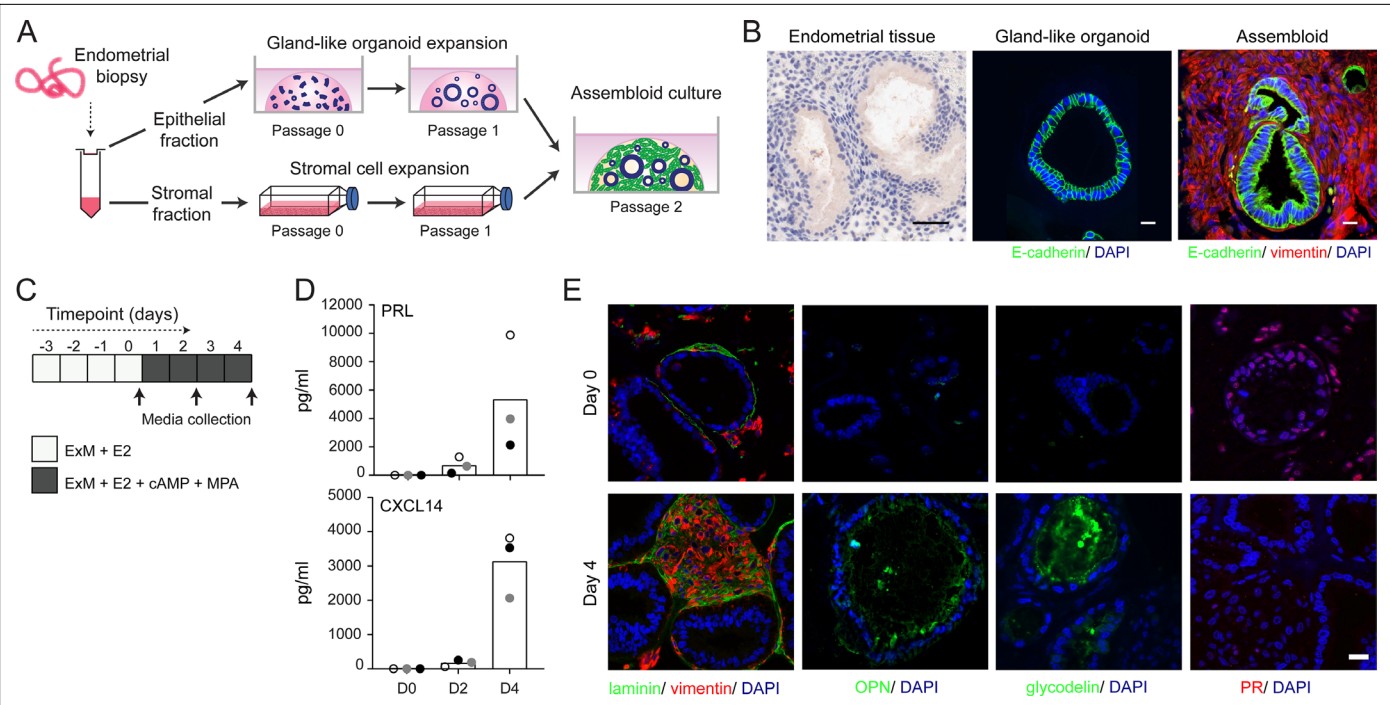

**Figure 1.** Establishment of endometrial assembloids. (**A**) Schematic for establishing endometrial assembloids. (**B**) Structural appearance of hematoxylin and eosin stained secretory endometrium, E-cadherin labelled gland-like organoids, and E-cadherin and vimentin stained endometrial assembloids. Scale bar = 50 μm. (**C**) Schematic summary of experimental design. (**D**) Secreted levels of PRL and CXCL14 were measured by ELISA in spent medium at the indicated timepoints. Data points are coloured to indicate secretion in assembloids established from different endometrial biopsies (n = 3). (**E**) Representative immunofluorescence labelling of laminin and vimentin, progesterone receptor (PR), glycodelin, and osteopontin (OPN) in undifferentiated (day 0, top panels) and decidualized (day 4; bottom panels) assembloids. Nuclei were counterstained with DAPI. Scale bar = 50 μm. ELISA data in (**B**) are available in *Figure 1—source data 1*.

The online version of this article includes the following figure supplement(s) for figure 1:

**Source data 1.** Secretion of PRL and CXCL14 by endometrial assembloids.

14 (CXCL14) (*Figure 1D*). Immunofluorescence microscopy provided further evidence that decidualizing assembloids mimic luteal phase endometrium, exemplified by laminin deposition by decidualizing EnSC, induction of osteopontin (*SPP1*) and accumulation of glycodelin (encoded by *PAEP*) in the lumen of secretory glands, and downregulation of the progesterone receptor (*PGR*) in both stromal and glandular compartments (*Figure 1E*).

We reasoned that once established assembloids may no longer require exogenous growth factors and pathway modulators for differentiation because of the presence of EnSC. To test this hypothesis, parallel gland-like organoids and assembloids were established from three endometrial biopsies and decidualized with E2, 8-bromo-cAMP, and MPA for 4 days in either expansion medium, base medium (*Supplementary file 1*: Table 1), or base medium with each exogenous factor added back individually. Induction of *PAEP* and *SPP1* was used to monitor the glandular differentiation response. As shown in *Figure 2*, differentiation of gland-like organoids in base medium markedly blunted the induction of *PAEP* and *SPP1* when compared to expansion medium. Add-back of individual factors did not restore the glandular response, with the exception of N-acetyl-L-cysteine (NAC). Addition of NAC at low concentration (1.25 mM) to base medium resulted in a robust glandular response in assembloids. Thus, in subsequent experiments assembloids were grown in expansion medium supplemented with E2 and then decidualized in minimal differentiation medium (MDM), consisting of base medium containing NAC, E2, 8-bromo-cAMP, and MPA.

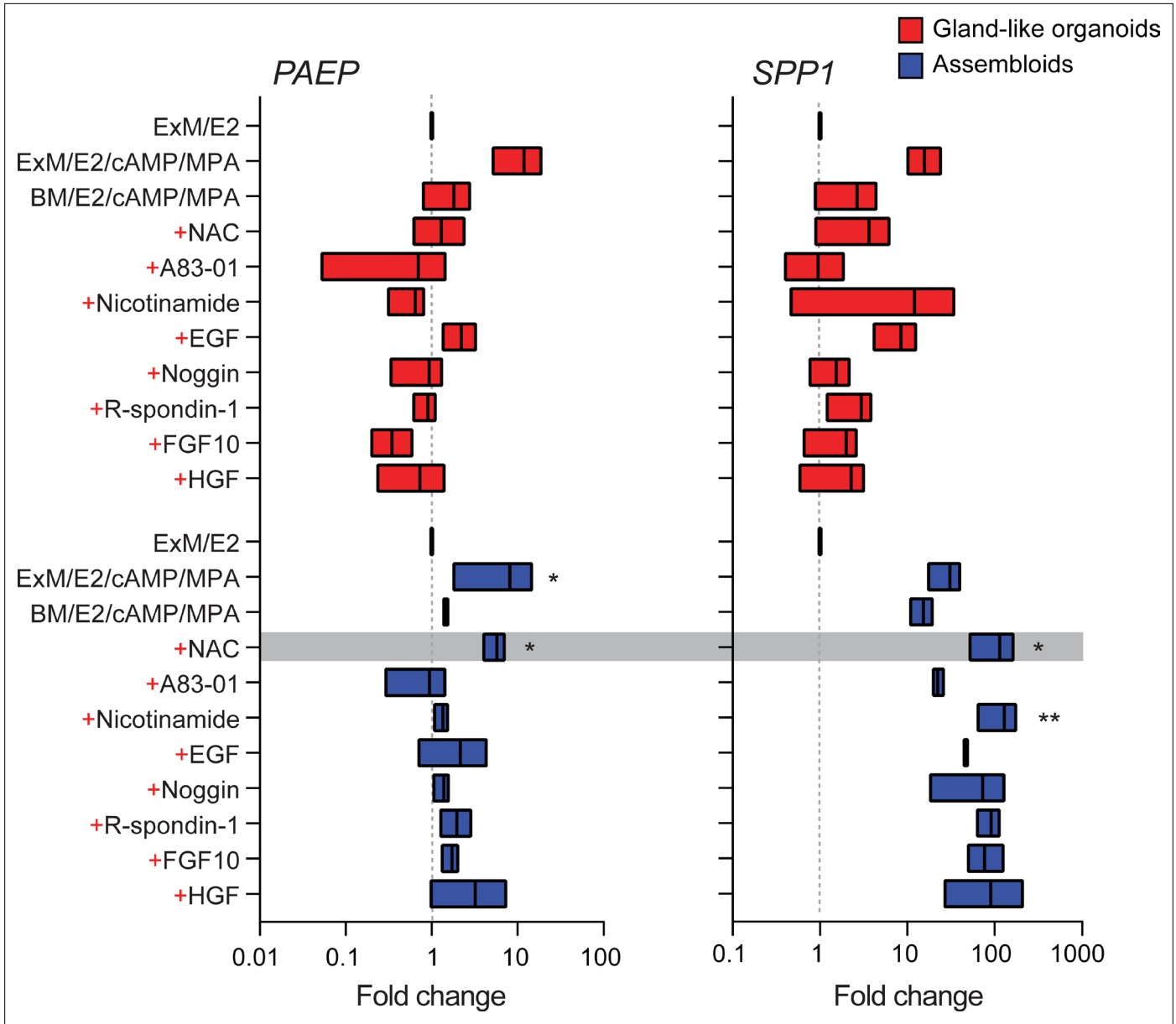

**Figure 2.** Characterization of a minimal differentiation medium for endometrial assembloids. Parallel gland-like organoids (red) and assembloids (blue) were established from three endometrial biopsies and decidualized with 8-bromo-cAMP and MPA for 4 days in either expansion medium (ExM), base medium (BM), or BM with each exogenous factor added back individually (+). Induction of *PAEP* and *SPP1* was used to monitor the glandular differentiation. The grey bar indicates the composition of the minimal differentiation medium selected for further use (BM supplemented with NAC, E2, cAMP, and MPA). Data are presented as fold-change relative to expression levels in undifferentiated organoids or assembloids cultured in ExM+ E2. Bars present minimal, maximal, and median fold-change. * and ** indicate p<0.05 and p<0.01 obtained by Friedman's test for matched samples. Relative expression values for biological replicates are available in *Figure 2—source data 1*.

The online version of this article includes the following figure supplement(s) for figure 2:

**Source data 1.** RTqPCR data associated with the minimal differentiation medium (MDM) experiments.

## Cellular complexity of decidualizing assembloids mimics midluteal endometrium

We hypothesized that, depending on the level of replicative stress (*Lucas et al., 2020*; *Abbas et al., 2019*), individual EpC and EnSC adopt distinct cellular states upon decidualization of endometrial assembloids. Based on previous time-course experiments in 2D cultures, we further speculated that divergence of cells into distinct subpopulations would be apparent by day 4 of differentiation (*Brosens*

*et al., 1999*; *Lucas et al., 2020*). To test this hypothesis, we performed single-cell RNA sequencing (scRNA-seq) on undifferentiated assembloids grown for 4 days in expansion medium and assembloids decidualized in MDM for four additional days (*Figure 3A*). Eleven distinct cell clusters were identified by Shared Nearest Neighbour (SNN) and Uniform Manifold Approximation and Projection (UMAP) analysis, segregating broadly into epithelial and stromal populations within the UMAP-1 dimension and into undifferentiated and differentiated subpopulations within the UMAP-2 dimension (*Figure 3B*). Each cell cluster was annotated based on expression of curated marker genes, which were cross-referenced with a publicly available data set (GEO: GSE4888) to determine their relative expression across the menstrual cycle in vivo (*Talbi et al., 2006*).

We identified five unambiguous EpC subsets. The glandular component of undifferentiated assembloids harboured actively dividing EpC (EpS1; n = 198) as well as EpC-expressing marker genes of E2-responsive proliferative phase endometrium (EpS2; n = 692), including *PGR* and *CPM* (*Figure 3C*). EpS3 (n = 29) consisted of ciliated EpC, expressing an abundance of genes involved in cilium assembly and organization, including *DNAI1* and *TUBA4B* (*Figure 3C*). Ciliated cells are the only glandular subpopulation present in both undifferentiated and decidualized assembloids. In vivo, EpS3 marker genes transiently peak during the early-luteal phase (*Figure 3C*). Decidualization of endometrial assembloids led to the emergence of two distinct EpC subsets, EpS4 (n = 434) and EpS5 (n = 208). Both clusters expressed canonical endometrial 'receptivity genes' (annotated in green in *Figure 3C*), that is, genes used in a clinical test to aid the timing of embryo transfer to the window of implantation in IVF patients (*Díaz-Gimeno et al., 2011*). In agreement, induction of EpS4 and EpS5 marker genes in vivo coincides with the transition from early- to midluteal phase. However, while expression of EpS4 marker genes, including *SOD2*, *MAOA*, and *PTGS1*, generally peaks during the midluteal window of implantation, EpS5 genes tend to persist or peak during the late-luteal phase (*Figure 3C*). Additional mining of the data revealed that transition from EpS4 to EpS5 coincides with induction of p16$^{INK4a}$ and p21$^{CIP1}$ in parallel with upregulation of 56 genes encoding secretory factors (*Figure 3—figure supplement 1*). Notably, several canonical implantation factors secreted by this subpopulation are also well-characterized SASP components, including dipeptidyl peptidase 4 (*DPP4*; *Kim et al., 2017*), growth differentiation factor 15 (*GDF15*; *Basisty et al., 2020*), and insulin-like growth factor binding protein 3 (*IGFBP3*; *Elzi et al., 2012*). Thus, EpS5 consists of senescent EpC producing an implantation-specific SASP.

Decidualized endometrial assembloids also harboured a sizable population of ambiguous cells expressing both epithelial and stromal genes (*Figure 3C* and *Figure 3—figure supplement 2*). A hallmark of this subset, termed 'transitional population' (TP; n = 472), is the induction of long non-coding RNAs involved in mesenchymal-epithelial and epithelial-mesenchymal transition (MET/EMT), such as *NEAT1* (nuclear paraspeckle assembly transcript 1) and *KCNQ1OT1* (KCNQ1 opposite strand/antisense transcript 1) (*Bian et al., 2019*; *Chen et al., 2021*). GO analysis showed that both EpS5 and the transitional population comprised secretory cells involved in ECM organization (*Figure 3D*). However, while EpS5 genes are implicated in neutrophil activation (a hallmark of premenstrual endometrium), genes expressed by the transitional population are uniquely enriched in GO terms such as 'wound healing', 'regulation of stem cell proliferation', 'blood coagulation', and 'blood vessel development' (*Figure 3D*), which points towards a putative role in tissue repair and regeneration.

The stromal fraction of undifferentiated assembloids consisted of actively dividing EnSC (stromal subpopulation 1 [SS1]; n = 434) and E2-responsive EnSC (SS2; n = 874) expressing proliferative phase marker genes, such as *PGR*, *MMP11*, and *CRABP2* (*Figure 3E*). As anticipated, decidualization of assembloids for 4 days led to a preponderance of pre-decidual cells (SS3; n = 495) as well as emerging decidual cells (SS4; n = 87) and senescent decidual cells (SS5; n = 118) (*Figure 3E*). Each of these subpopulations expressed marker genes identified previously by scRNA-seq reconstruction of the decidual pathway in standard primary EnSC cultures (*Lucas et al., 2020*). Pre-decidual cells in SS3 express *HAND2*, a key decidual transcription factor (*Marinić et al., 2021*), as well as previously identified genes encoding secreted factors, including *VEGFA* (vascular endothelial growth factor A), *CRISPLD2* (a progesterone-dependent anti-inflammatory response gene coding cysteine-rich secretory protein LCCL domain containing 2), *IL15* (interleukin 15), and *TIMP3* (TIMP metallopeptidase inhibitor 3) (*Lucas et al., 2020*). Novel candidate pre-decidual genes were also identified, such as *DDIT4* (DNA damage-inducible transcript 4), encoding a stress response protein intimately involved in autophagy, stemness, and antioxidative defences (*Ho et al., 2020*; *Miller et al., 2020*). Decidual

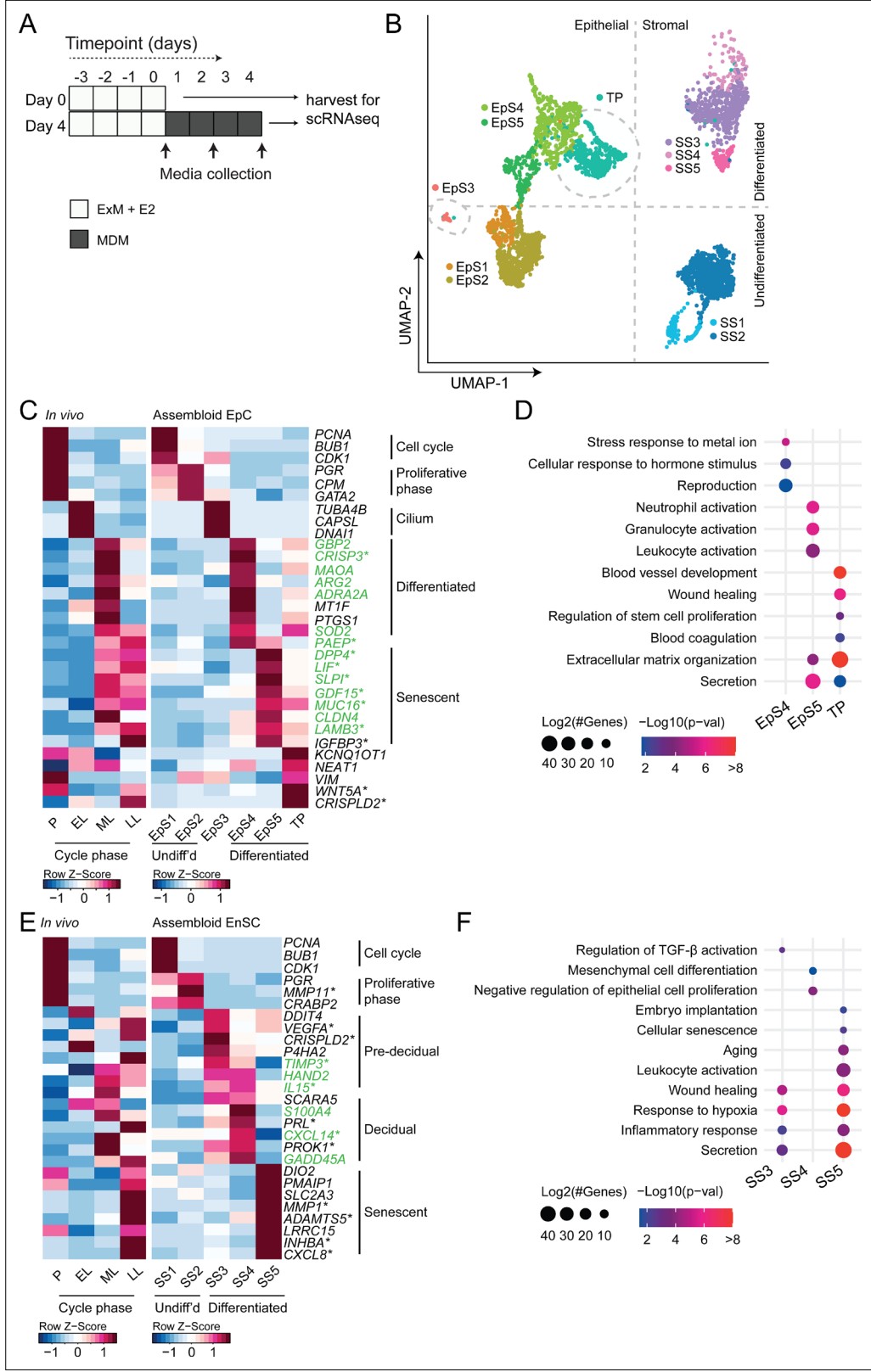

**Figure 3.** Decidualizing assembloids mimic midluteal endometrium. (**A**) Schematic overview of experimental design. ExM: expansion medium; MDM: minimal differentiation medium. (**B**) Uniform Manifold Approximation and Projection (UMAP) visualizing epithelial and stromal subsets (EpS and SS, respectively) identified by single-cell transcriptomic analysis of undifferentiated and decidualized assembloids. A transitional population (TP) consisting

*Figure 3 continued*

of cells expressing epithelial and stromal markers is also shown. Dotted lines indicate the separation of EpS and SS in UMAP_1 and of undifferentiated and differentiated subpopulations in UMAP_2. Dotted circles indicate ciliated (EpS3) and TP, which did not fit these broad segregations. (**C**) Composite heatmaps showing relative expression (Z-scores) of epithelial marker genes across the menstrual cycle in vivo and in undifferentiated and decidualized assembloids. Highlighted in green are genes that mark the midluteal window of implantation (*Díaz-Gimeno et al., 2011*), whereas genes encoding secreted proteins are indicated by * (*Uhlén et al., 2015*). See also *Figure 3— figure supplement 1*. (**D**) Dot plots showing GO terms related to biological processes enriched in different epithelial populations in decidualizing assembloids. The dot size represents the number of genes in each GO term and the colour indicates FDR-corrected p-value. (**E**) Composite heatmaps showing relative expression (Z-scores) of stromal marker genes across the menstrual cycle in vivo and in undifferentiated and decidualized assembloids. Highlighted in green are genes that mark the midluteal window of implantation (*Díaz-Gimeno et al., 2011*), whereas genes encoding secreted proteins are indicated by * (*Uhlén et al., 2015*). (**F**) Dot plots showing GO terms related to biological processes enriched in different stromal subpopulations in decidualizing assembloids. See also *Figure 3—figure supplements 1 and 2* and 3. Complete epithelial subpopulation marker lists can be found in *Figure 3—source data 1*. GO analysis outputs can be found in . Complete stromal subpopulation marker lists can be found in .

The online version of this article includes the following figure supplement(s) for figure 3:

**Source data 1.** Epithelial subpopulation markers.

**Source data 2.** GO analysis of differentiated subpopulations.

**Source data 3.** Stromal sub-population markers.

**Figure supplement 1.** Heatmap showing relative expression (Z-scores) of genes encoding the cyclin-dependent kinase inhibitors p16$^{INK4a}$ and p21$^{CIP1}$ as well as SASP-related genes in epithelial and transitional subpopulations in decidualizing assembloids.

**Figure supplement 2.** Heatmap showing relative expression (Z-scores) of epithelial-mesenchymal transition/mesenchymal-epithelial transition (EMT/MET), epithelial and mesenchymal marker genes in the transitional population (TP), epithelial (EpS4-5) and stromal (SS3-5) subpopulations in decidualizing assembloids.

**Figure supplement 3.** Heatmap showing relative expression (Z-scores) of genes encoding the cyclin-dependent kinase inhibitors p16$^{INK4a}$ and p21$^{CIP1}$ as well as secretory and SASP-related genes in stromal subpopulations (SS3-5) in decidualizing assembloids.

---

cells (SS4) and senescent decidual cells (SS5) express *SCARA5* and *DIO2*, respectively (*Figure 3E*), two stroma-specific marker genes identified by scRNA-seq analysis of mid- and late-luteal endometrial biopsies (*Lucas et al., 2020*). SS3 and SS4 genes mapped to the early- and midluteal phase of the cycle, whereas SS5 genes peak in the late-luteal phase, that is, prior to menstrual breakdown. Notably, the transcriptomic profiles of SS3 and SS5 are enriched in GO terms such as 'Wound healing', 'Response to hypoxia', and 'Inflammatory response', suggesting that both clusters comprise stressed cells (*Figure 3F*). However, the nature of the cellular stress response differs between these populations with only senescent decidual cells (SS5) expressing genes enriched in categories such as 'Embryo implantation', 'Cellular senescence', 'Aging', and 'Leukocyte activation'. By contrast, few notable categories were selectively enriched in decidual cells (e.g. 'Mesenchymal cell differentiation'), rendering the lack of GO terms that pertain to stress, inflammation, or wound healing perhaps the most striking observation. In keeping with the GO analysis, senescent decidual cells (SS5) express a multitude of SASP-related genes (*Figure 3—figure supplement 3*), including matrix metallopeptidases (e.g. *MMP3, 7, 9, 10, 11,* and *14*), insulin-like growth factor binding proteins (e.g. *IGFBP1, 3, 6,* and *7*), growth factors (e.g. *AREG, FGF2, FGF7, HGF,* and *VEGFA*) and growth factor receptors (*PDGFRA* and *PDGFRB*), cytokines (e.g. *LIF, IL6, IL1A,* and *IL11*), chemokines (e.g. *CXCL8* and *CXCL1*), and members of the TGF-β superfamily of proteins (e.g. *GDF15, INHBA,* and *BMP2*). By contrast, decidual cells are characterized by expression of a unique network of secretory genes, some encoding ECM proteins (e.g. *COL1A1, COL3A1,* and *LAMA4*) and other known decidual markers (e.g. *PRL, PROK1,* and *WNT4*) as well as factors involved in uNK cell chemotaxis and activation (e.g. *CCL2, CXCL14,* and *IL15*) (*Figure 3—figure supplement 3*).

Taken together, single-cell analysis of undifferentiated and decidualized assembloids revealed a surprising level of cellular complexity. Each epithelial and stromal subpopulation appears functionally distinct and maps to a specific phase of the menstrual cycle. Transition between cellular states is

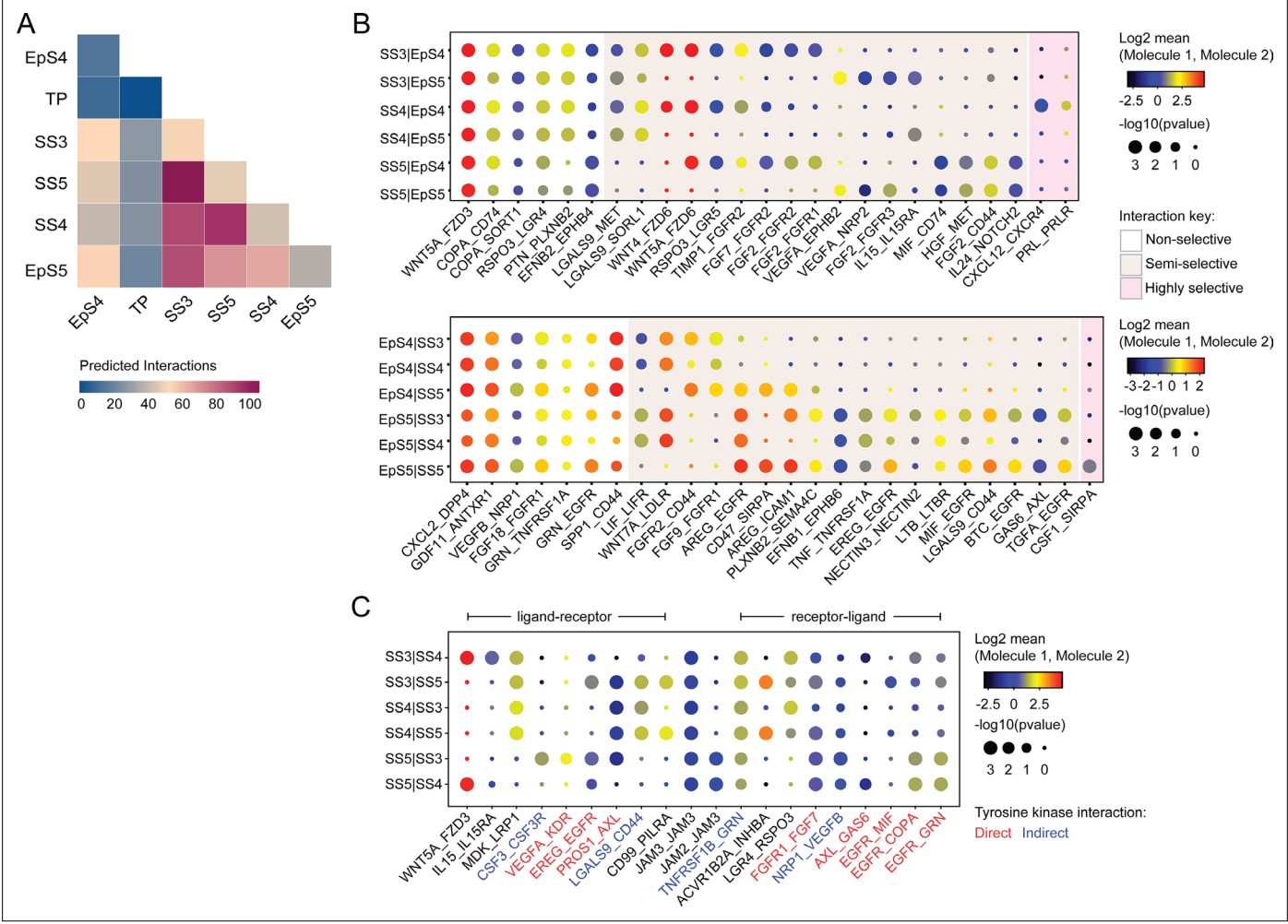

**Figure 4.** Putative receptor-ligand interactions in decidualizing assembloids. (**A**) Heatmap showing the total number of cell-cell interactions predicted by CellPhoneDB between different subpopulations in decidualizing assembloids. (**B**) Dot plots of representative ligand-receptor interactions between stromal subsets (SS) and epithelial subsets (EpS) (upper panel) and EpS and SS (lower panel) in decidualizing assembloids. Circle size and colour indicate p-value and the means of the average expression value of the interacting molecules, respectively. Shaded boxes were used to group putative interactions by level of selectivity. (**C**) Dot plot of representative ligand-receptor and receptor-ligand interactions between stromal subpopulations in decidualizing assembloids. Direct and indirect tyrosine kinase interactions are indicated by red and blue labels, respectively. Complete tables of predicted ligand-receptor interactions can be found in *Figure 4—source data 1*.

The online version of this article includes the following figure supplement(s) for figure 4:

**Source data 1.** CellPhoneDB prediction of cell-cell interactions.

predicated on changes in cell cycle status, ranging from actively dividing cells in proliferating assembloids to the emergence upon differentiation of highly secretory senescent epithelial and decidual subpopulations, resembling premenstrual endometrium. However, the dominant subpopulations on day 4 of decidualization are EpS4 and SS3, which map to the midluteal implantation window in vivo.

## Receptor-ligand interactions in decidualizing assembloids

We used CellPhoneDB, a publicly available online repository of highly curated receptor-ligand interactions, to explore putative interactions between subpopulations in decidualizing assembloids. This computational tool also takes into account the subunit architecture of both ligands and receptors in heteromeric complexes (*Efremova et al., 2020*; *Vento-Tormo et al., 2018*). The number of predicted interactions is depicted in *Figure 4A*, showing a conspicuous lack of crosstalk between the transitional population and any other populations. Conversely, the most abundant interactions centre around the secretory subpopulations, EpS5 and SS5.

A total of 270 significantly enriched (non-integrin) receptor-ligand interactions (FDR-corrected p<0.05) were identified between epithelial and stromal subsets in decidualizing assembloids (*Figure 4—source data 1*), a representative selection of which are shown in *Figure 4B*. Within the multitude of predicted complex interactions, three broad categories can be discerned. First, there are non-selective interactions involving ligands produced by all subpopulations in one compartment acting on receptors expressed by all subsets in the other compartment. Second, there are semi-selective stromal-epithelial interactions involving three or four subpopulations across both compartments. For example, binding of WNT5A secreted by all decidual stromal subsets to FZD3 (frizzled class receptor 3) expressed on all EpCs represents a non-selective receptor-ligand interaction, whereas binding of WNT5A or WNT4 to FZD6 is a predicted semi-selective interaction, involving all stromal subsets (SS3-5) and EpS4 but not EpS5 (*Figure 4B*). While FZD3 activates the canonical β-catenin pathway, FZD6 functions as a negative regulator of this signalling cascade (*Corda and Sala, 2017*). Finally, we identified only three highly selective receptor-ligand interactions (*Figure 4B*), two of which involved secretion of decidual ligands, prolactin (PRL) and C-X-C motif chemokine ligand 12 (CXCL12), acting on their cognate receptors expressed on receptive EpC (EpS4). CXCL12-dependent activation of C-X-C motif chemokine receptor 4 (CXCR4) has been shown to promote motility of EpC (*Zheng et al., 2020*), whereas PRL is a lactogenic hormone that stimulates glandular secretion in early pregnancy (*Burton et al., 2020*).

In contrast to stromal-epithelial communication, non-selective interactions are predicted to be rare between decidual subsets. Instead, communication appears governed largely by a combinatorial network of receptor-ligand interactions (*Figure 4C*). For example, colony stimulating factor 3 (CSF3) and vascular endothelial growth factor A (VEGFA) produced by senescent decidual cells (SS5) are predicted to impact selectively on pre-decidual cells (SS3), whereas secretion of inhibin A (INHBA) may engage both pre-decidual and decidual cells (SS4). Other interactions are predicted to govern crosstalk between SS3 and SS4, such as modulation of the WNT pathway in response to binding of R-spondin 3 (RSPO3) to leucine-rich repeat-containing G protein-coupled receptor 4 (LGR4). A striking observation is the overrepresentation of receptor tyrosine kinases implicated in SS3 and SS5 signal transduction as well as the involvement of receptors that signal through downstream cytoplasmic tyrosine kinases, including CSF3 receptor (CSF3R) and CD44 (*Figure 4C*; *Corey et al., 1998*; *van der Voort et al., 1999*).

## Tyrosine kinase-dependent stress responses determine the fate of decidual cells

The CellPhoneDB analysis inferred that epithelial-stromal crosstalk in assembloids is robust, buffered by numerous non-selective interactions, whereas decidual subsets are reliant on selective receptor-ligand interactions and activation of distinct signal transduction pathways. For example, the predicted tyrosine kinase dependency of pre-decidual (SS3) and senescent decidual cells (SS5) raised the possibility that these subpopulations can be targeted by tyrosine kinase inhibitors, such as dasatinib (*Brighton et al., 2017*; *Zhu et al., 2015*), a second-generation, broad-spectrum ATP-competitive protein tyrosine kinase inhibitor (*Aguilera and Tsimberidou, 2009*; *Li et al., 2010*). To test this supposition, we generated single-cell transcriptomic profiles of assembloids decidualized for 4 days in the presence of dasatinib (*Figure 5A*). We found that decidualization in the presence of dasatinib had a dramatic impact on stromal subpopulations, virtually eliminating senescent decidual cells (SS5, n = 7) and increasing the abundance of decidual cells ninefold (SS4, n = 882; *Figure 5B*). Apart from a modest reduction in pre-decidual cells (SS3), dasatinib also impacted markedly on transitional cells, reducing their numbers by 76% . By contrast, the effect on epithelial populations was confined to a modest reduction in senescent EpC (EpS5) (*Figure 5B*). Further, relatively few genes were perturbed significantly (FDR-corrected p<0.05) upon dasatinib treatment in epithelial populations (*Figure 5C*). In the stroma, dasatinib triggered a conspicuous transcriptional response in pre-decidual (SS3) and transitional cells, whereas gene expression in decidual cells (SS4) and the few remaining senescent decidual cells (SS5) was largely unaffected (*Figure 5C*). In transitional cells, dasatinib simultaneously upregulated genes encoding canonical mesenchymal markers (e.g. *SNAI2*, *TWIST2*, *ZEB1*, *COL1A1*, and *FBN1*; *Owusu-Akyaw et al., 2019*) and decidual factors (e.g. *SCARA5*, *FOXO1*, *GADD45A*, *IL15*, *CXCL14*, and *SGK1*; *Gellersen and Brosens, 2014*), suggesting that MET accounts for the emergence of this population upon decidualization (*Figure 5—source data 1*). In pre-decidual cells,

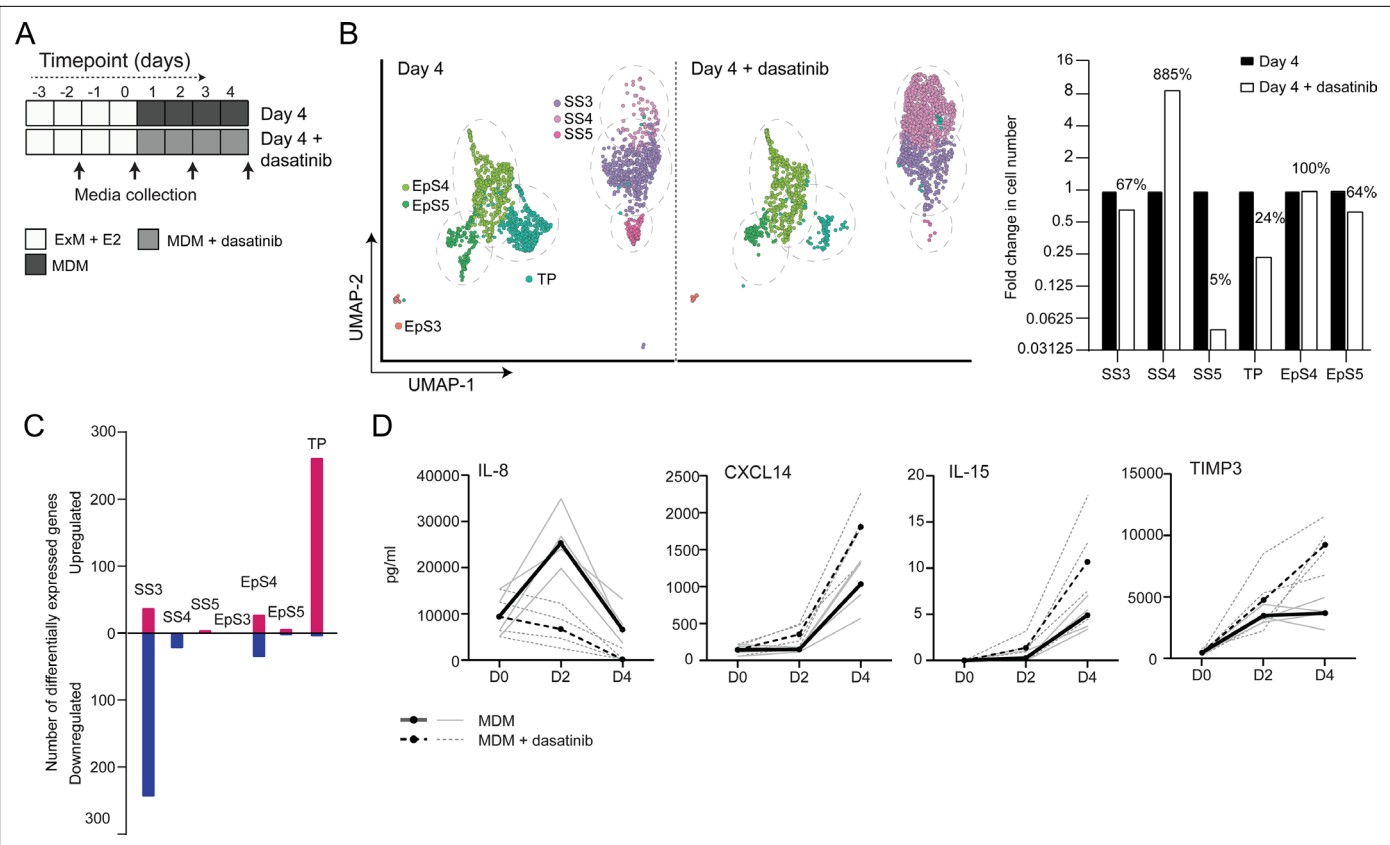

**Figure 5.** Tyrosine kinase-dependent stress responses determine the fate of decidual cells. (**A**) Schematic overview of experimental design. ExM: expansion medium; MDM: minimal differentiation medium. (**B**) Uniform Manifold Approximation and Projection (UMAP) visualization (left panel) and relative proportions (right panel) of subpopulations in endometrial assembloid decidualized in the presence or absence of dasatinib. (**C**) Number of differentially expressed genes (DEGs) in each subpopulation in response to dasatinib pre-treatment. (**D**) Secreted levels of CXCL8 and decidual cell factors in spent medium from assembloids treated with or without dasatinib. Secreted levels in individual assembloids established from four different endometrial assembloids decidualized with or without dasatinib are shown by dotted and solid lines, respectively. Full lists of DEGs and associated GO analysis can be found in *Figure 5—source data 1* and *Figure 5—source data 2*, respectively. Data used in (**D**) are available in *Figure 5—source data 3*.

The online version of this article includes the following figure supplement(s) for figure 5:

**Source data 1.** Differentially expressed genes for day 4 populations treated with and without dasatinib.

**Source data 2.** GO analysis for day 4 populations treated with and without dasatinib.

**Source data 3.** ELISA data.

dasatinib inhibited the expression of a network of genes enriched in GO categories such as 'Response to wounding' (FDR-corrected $p=3.5 \times 10^{-5}$), 'Response to stress' (FDR-corrected $p=3.8 \times 10^{-5}$), and 'Response to oxidative stress' (FDR-corrected $p=1.3 \times 10^{-4}$), indicative of a blunted stress response. To substantiate this finding, we measured the secreted levels of CXCL8 (IL-8), a potent inflammatory mediator implicated in autocrine/paracrine propagation of cellular senescence (*Acosta et al., 2008*; *Kuilman et al., 2008*), in assembloids decidualized with or without dasatinib. CXCL14, IL-15, and TIMP3 levels were also measured to monitor the decidual response. As shown in *Figure 5D*, dasatinib completely abrogated the release of CXCL8 by pre-decidual cells while markedly enhancing subsequent secretion of CXCL14, IL-15, and TIMP3, which are involved in effecting immune clearance of senescent decidual cells (*Brighton et al., 2017*; *Lucas et al., 2020*; *Kong et al., 2021*). Together, these observations not only support the CellPhoneDB predictions but also indicate that the amplitude of the cellular stress response during the pre-decidual phase determines the subsequent decidual trajectory, with low levels accelerating differentiation and high levels promoting cellular senescence and MET.

# Modelling the impact of decidual subpopulations on human embryos

We postulated that decidual invasion by human embryos that have breached the luminal endometrial epithelium depends on an acute cellular senescence and transient SASP production, rich in growth factors and proteases. Conversely, we reasoned that lack of senescent decidual cells or unconstrained SASP should simulate pathological implantation environments associated with implantation failure and early pregnancy loss, respectively. To test this hypothesis, we constructed a simple implantation

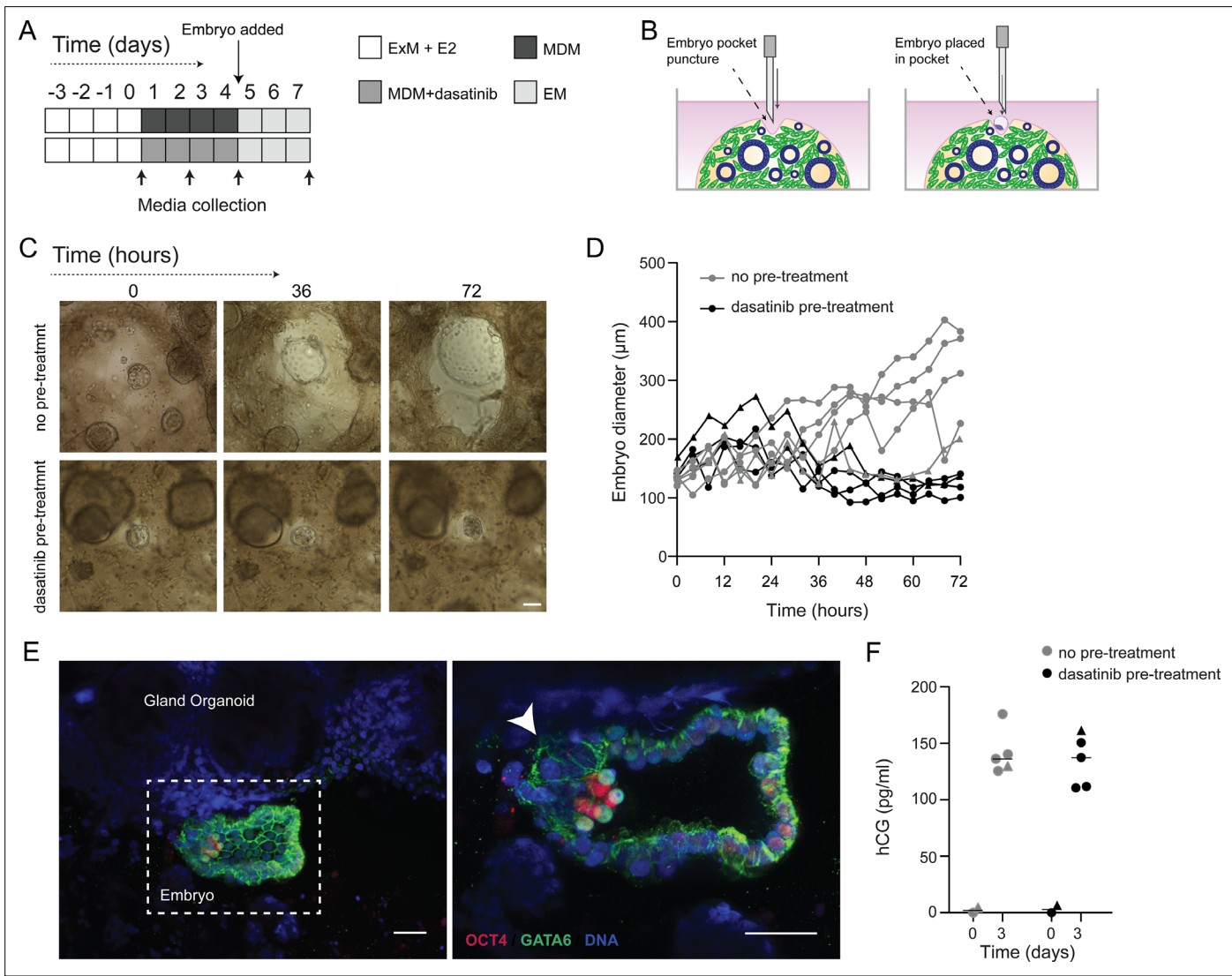

**Figure 6.** Impact of decidual senescence in assembloids on co-cultured human blastocysts. (**A**) Diagram showing experimental design. ExM: expansion medium; MDM: minimal differentiation medium; EM: embryo medium. (**B**) Schematic drawing of co-culture method. (**C**) Representative time-lapse images of blastocysts embedded in assembloids following decidualization for 96 hr in the absence (upper panels) or presence (lower panels) of dasatinib. Scale bar = 100 μm. See also *Figure 6—figure supplement 1*. (**D**) Embryo diameters (μm) measured over 72 hr when embedded in decidualizing assembloids pre-treated with or without dasatinib. (**E**) OCT4 and GATA6 immunofluorescence marking the epiblast and hypoblast, respectively, in a blastocyst attached by proliferating polar trophectoderm (arrowhead) to decidual assembloids. Scale bar = 50 μM. (**F**) Secreted levels of human chorionic gonadotropin (hCG) in blastocyst-endometrial assembloid co-cultures. Individual embryo diameter measurements for biological replicates in (**D**) are available in *Figure 6—source data 1*. Individual ELISA data used in (**F**) are available in *Figure 6—source data 2*.

The online version of this article includes the following figure supplement(s) for figure 6:

**Source data 1.** Embryo expansion measurement.

**Source data 2.** Embryo human chorionic gonadotropin (hCG) secretion.

**Figure supplement 1.** Stromal migration towards the polar trophectoderm of expanding embryos in differentiated endometrial assembloids.

**Figure supplement 2.** Dasatinib prevents disintegration of decidualizing assembloids.

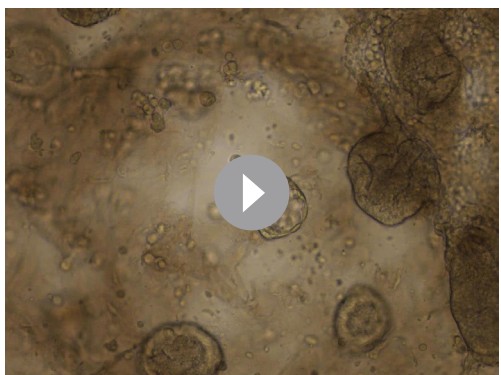

**Video 1.** Time-lapse microscopy of a human blastocyst embedded in a decidualizing assembloid. Representative video of a human blastocyst embedded in an assembloid, as imaged by time-lapse microscopy over 72 hr with images captured every 60 min.
https://elifesciences.org/articles/69603/figures#video1

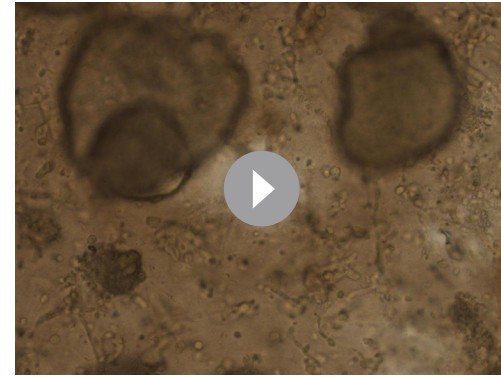

**Video 2.** Time-lapse microscopy of a human blastocyst embedded in a decidualizing assembloid pre-treated with dasatinib. Representative video of a human blastocyst embedded in an assembloid which had been pre-treated with dasatinib, as imaged by time-lapse microscopy over 72 hr with images captured every 60 min.
https://elifesciences.org/articles/69603/figures#video2

model by embedding human embryos in endometrial assembloids. To this end, assembloids were first decidualized for 96 hr in the presence or absence of dasatinib, washed and cultured in embryo medium, consisting of MDM with added supplements (*Figure 6A* and *Supplementary file 1*: Table 1). Day 5 human blastocysts were placed into small pockets created in the decidualized assembloids (*Figure 6B*), one embryo per assembloid, and individual co-cultures imaged using time-lapse microscopy over 72 hr. Co-cultured blastocysts (n = 5) expanded markedly when placed in decidualized assembloids that were not pre-treated with dasatinib (*Figure 6C and D*). Time-lapse microscopy revealed intense cellular movement in the stromal compartment as well as evidence that interaction between migratory decidual cells and polar trophectoderm promotes adherence and early invasion of the embryo (SI *Video 1* and *Figure 6—figure supplement 1*). Retrieval and processing of one attached embryo demonstrated proliferating polar trophectoderm and expression of OCT4 and GATA6 in the epiblast and hypoblast, respectively (*Figure 6E*). A major limitation of this implantation model is that persistence of senescent decidual cells also causes gradual disintegration of the assembloids (*Figure 6—figure supplement 2*). By contrast, pre-treatment with dasatinib, which accelerates decidualization and all but eliminates decidual senescence, resulted in much more robust assembloids. However, all embedded blastocysts (n = 5) failed to expand in this model (*Figure 6C and D*). Further, movement of the decidual matrix was greatly reduced and directed migration or attachment of decidual cells to the blastocyst was not observed (SI *Video 2*). Secreted levels of human chorionic gonadotropin (hCG) did not differ between co-cultures (*Figure 6E*), suggesting that all embryos remained viable over the 72 hr observation period. Thus, while our experimental design precluded modelling of physiological embryo implantation, aspects of different pathological endometrial states underlying reproductive failure, that is, implantation failure and miscarriage, were recapitulated in assembloids.

## Discussion

Here we report on the development of endometrial assembloids, consisting of gland-like organoids surrounded by a matrix rich in primary EnSC, as novel model to parse the cellular dynamics that govern embryo implantation in cycling human endometrium. While assembloids complement and advance other recently described endometrial organoid models (*Boretto et al., 2017*; *Cheung et al., 2021*; *Fitzgerald et al., 2019*; *Luddi et al., 2020*; *Turco et al., 2017*), they still lack the cellular complexity of native endometrium, including uNK cells, macrophages, and vascular cells. Nevertheless, we demonstrated that aspects of pathological implantation events can be recapitulated in assembloids, rendering them useful as novel models to study mechanisms of reproductive failure and evaluate potential therapeutic interventions.

Single-cell analysis of differentiating endometrial assembloids indicates that the sequence of events leading up to the implantation window, and beyond, requires divergence of both glandular EpC and EnSC into differentiated and senescent subpopulations, a process likely determined by the level of replication stress incurred by individual cells in the preceding proliferative phase (*Brighton et al., 2017*). Importantly, we demonstrate that acute senescence in glandular EpC (EpS5) underpins production of an implantation-specific SASP, comprising canonical implantation factors and growth factors, such as amphiregulin (*AREG*) and epiregulin (*EREG*), implicated in transforming cytotrophoblasts into extravillous trophoblasts (*Cui et al., 2020*; *Yu et al., 2019*). On the other hand, the transcriptome profile of differentiated EpC (EpS4) revealed a pivotal role for this subpopulation in prostaglandin and glycodelin synthesis. Prostaglandins, and specifically PGE2, are indispensable for implantation (*Ruan et al., 2012*), whereas glycodelin is an abundantly secreted, multifaceted glycoprotein involved in blastocyst attachment, trophoblast differentiation, and immune modulation in early pregnancy (*Lee et al., 2016*). Further, differentiated EpC highly express *SLC2A1*, encoding the major glucose transporter GLUT1. Glucose is required for glycogen synthesis, an essential component of glandular secretions that nourishes the conceptus prior to the onset of placental perfusion around 10 weeks of pregnancy (*Burton et al., 2020*). The fate and function of senescent EpC in pregnancy are unknown. Arguably, localized secretion of proteinases by senescent EpC may promote breakdown of the surrounding basement membrane, thereby facilitating endoglandular trophoblast invasion and access to histotrophic nutrition in early gestation (*Huppertz, 2019*; *Moser et al., 2010*). In non-conception cycles, the abundance of p16$^{INK4}$-positive glandular EpC rises markedly during the late-luteal phase (*Brighton et al., 2017*), indicating that senescent EpC are progesterone-independent and likely responsible for glandular breakdown in the superficial endometrial layer at menstruation.

Decidual transformation of EnSC in assembloids unfolded largely as anticipated from previous studies, that is, starting with an acute pre-decidual stress response and leading to the emergence of both decidual and senescent decidual subpopulations (*Brighton et al., 2017*; *Lucas et al., 2020*; *Kong et al., 2021*). Like their epithelial counterparts, senescent decidual cells have a conspicuous secretory phenotype. We identified 56 and 72 genes encoding secreted factors upregulated in senescent epithelial and decidual subpopulations, respectively. However, only 15 genes were shared, indicating that the SASP generated in both cellular compartments is distinct. As glandular secretions drain into the uterine cavity, the embryonic microenvironment is therefore predicted to change abruptly upon breaching of the luminal epithelium. Recent comparative metabolomics of apical and basolateral endometrial gland-like organoid secretomes also supports the prediction of an asymmetrical profile of glandular secretions in the pre- and post-implantation microenvironments (*Simintiras et al., 2021*).

Based on computational predictions of ligand-receptor interactions, we demonstrated that the decidual response in assembloids can be targeted pharmacologically with only modest impact on glandular function and, by extension, the preimplantation embryo milieu. Specifically, dasatinib, a tyrosine kinase inhibitor, was highly effective in blunting the pre-decidual stress response, leading to a dramatic expansion of anti-inflammatory decidual cells and near-total elimination of senescent decidual cells. Dasatinib also inhibited the emergence of TP and shifted the transcriptional profile of the remaining transitional cells towards a decidual phenotype. An analogous population of ambiguous cells expressing both epithelial and mesenchymal marker genes was recently identified in midluteal endometrium by scRNA-seq analysis (*Lucas et al., 2020*). Further, based on CellPhoneDB and GO analyses, transitional cells are predicted to be highly autonomous and involved in tissue regeneration, in line with experimental evidence that MET drives re-epithelization of the endometrium following menstruation and parturition (*Owusu-Akyaw et al., 2019*; *Patterson et al., 2013*). Thus, the level of endogenous cellular stress generated by the endometrium during the window of implantation calibrates the subsequent decidual trajectory, either promoting the formation of a robust decidual matrix or facilitating tissue breakdown and repair. Further, an in-built feature of both trajectories is self-enforcement as decidual cells recruit and activate uNK cells to eliminate their senescent counterparts (*Brighton et al., 2017*; *Lucas et al., 2020*; *Kong et al., 2021*), whereas senescent decidual cells induce secondary senescence in neighbouring decidual (*Brighton et al., 2017*; *Ozaki et al., 2017*) and, plausibly, uNK cells (*Rajagopalan and Long, 2012*).

Clinically, recurrent pregnancy loss, defined as multiple miscarriages, is associated with loss of endometrial clonogenicity (*Lucas et al., 2016*; *Diniz-da-Costa et al., 2021*), uNK cell deficiency and

excessive decidual senescence (*Lucas et al., 2020*; *Tewary et al., 2020*), and rapid conceptions (also referred to as 'superfertility') (*Dimitriadis et al., 2020*; *Ticconi et al., 2020*). Conversely, lack of a proliferative gene signature in midluteal endometrium and premature expression of decidual PRL have been linked to recurrent implantation failure (*Berkhout et al., 2020b*; *Koler et al., 2009*; *Koot et al., 2016*), a pathological condition defined by a failure to achieve a pregnancy following transfer of one or more high-quality embryos in multiple IVF cycles (*Polanski et al., 2014*). We reasoned that these aberrant implantation environments can be recapitulated in assembloids by manipulating the level of decidual senescence. In line with these predictions, the presence of senescent decidual cells created a permissive environment in which migratory decidual cells interacted with expanding blastocysts, although continuous SASP production also promoted breakdown of the assembloids. Conversely, in the absence of senescent decidual cells, non-expanding embryos became entrapped in a robust but stagnant decidual matrix. These observations are in keeping with previous studies demonstrating that implantation of human embryos depends critically on the invasive and migratory capacities of decidual cells (*Berkhout et al., 2020a*; *Gellersen et al., 2010*; *Grewal et al., 2008*; *Weimar et al., 2012*). Our co-culture experiments also highlighted the shortcomings of assembloids as an implantation model, including the lack of a surface epithelium to create distinct pre- and post-implantation microenvironments and the absence of key cellular constituents, such as innate immune cells.

In summary, parsing the mechanisms that control implantation has been hampered by the overwhelming complexity of factors involved in endometrial receptivity. Our single-cell analysis of decidualizing assembloids suggests that this complexity reflects the reliance of the human endometrium on rapid E2-dependent proliferation and replicative exhaustion to generate both differentiated and senescent epithelial and stromal subpopulations in response to the postovulatory rise in progesterone. We demonstrate that senescent cells in both cellular compartments produce distinct bioactive secretomes, which plausibly prime pre-implantation embryos for interaction with the luminal epithelium and then stimulate encapsulation by underlying decidual stromal cells. Based on our co-culture observations, we predict that a blunted pre-decidual stress response causes implantation failure because of a lack of senescence-induced tissue remodelling and accelerated decidualization. Conversely, a heightened stress response leading to excessive decidual senescence may render embryo implantation effortless, albeit in a decidual matrix destined for breakdown and repair. Finally, we demonstrated that pre-decidual stress responses can be modulated pharmacologically, highlighting the potential of endometrial assembloids as a versatile system to evaluate new or repurposed drugs aimed at preventing reproductive failure.

## Materials and methods
### Ethical approvals, endometrial samples, and human blastocysts
Endometrial biopsies were obtained from women attending the Implantation Research Clinic, University Hospitals Coventry and Warwickshire National Health Service Trust. Written informed consent was obtained in accordance with the Declaration of Helsinki 2000. The study was approved by the NHS National Research Ethics Committee of Hammersmith and Queen Charlotte's Hospital NHS Trust (1997/5065) and Tommy's Reproductive Health Biobank (Project TSR19-002E, REC Reference: 18/WA/0356). Timed endometrial biopsies were obtained 6–11 days after the post-ovulatory LH surge using a Wallach Endocell Endometrial Cell Sampler. Patient demographics for the samples used in each experiment are detailed in *Supplementary file 1*: Table 2.

The use of vitrified human blastocysts was carried out under a Human Fertilisation and Embryology Authority research licence (HFEA: R0155) with local National Health Service Research Ethics Committee approval (04/Q2802/26). Spare blastocysts were donated to research following informed consent by couples who had completed their fertility treatment at the Centre for Reproductive Medicine, University Hospitals Coventry and Warwickshire National Health Service Trust. Briefly, women underwent ovarian stimulation and oocytes were collected by transvaginal ultrasound-guided aspiration and inseminated with prepared sperm (day 0). All oocytes examined 16–18 hr after insemination and classified as normally fertilized were incubated under oil in 20–25 µl drops of culture media (ORIGIO Sequential Cleav and Blast media, CooperSurgical, Denmark) at 5% $O_2$, 6% $CO_2$, 89% $N_2$ at 37 °C. Following culture to day 5 of development, the embryo(s) with the highest quality was selected for transfer, whereas surplus embryos considered top-quality blastocysts were cryopreserved

on day 5 or 6 by vitrification using Kitazato vitrification media (Dibimed, Spain) and stored in liquid nitrogen. Prior to their use in the co-culture, vitrified blastocysts were warmed using the Kitazato vitrification warming media (Dibimed, Spain) and underwent zona pellucida removal using a Saturn 5 Laser (CooperSurgical). Blastocysts were then incubated for 1 hr under oil in 20 µl drops of culture media (ORIGIO Sequential Blast media, CooperSurgical) at 5% $O_2$, 6% $CO_2$, 89% $N_2$ at 37 °C and allowed to re-expand.

## Processing of endometrial biopsies and primary EnSC cultures

Unless otherwise stated, reagents were obtained from Life Technologies (Paisley, UK). Cell cultures were incubated at 37 °C, 5% $CO_2$ in a humidified incubator. Centrifugation and incubation steps were performed at room temperature unless stated otherwise. Fresh endometrial biopsies were processed as described previously (*Barros et al., 2016*). Briefly, tissue was finely minced for 5 min using a scalpel blade. Minced tissue was then digested enzymatically with 0.5 mg/ml collagenase I (Sigma-Aldrich, Gillingham, UK) and 0.1 mg/ml deoxyribonuclease (DNase) type I (Lorne Laboratories, Reading, UK) in 5 ml phenol red-free Dulbecco's Modified Eagle Medium (DMEM)/F12 for 1 hr at 37 °C, with regular vigorous shaking. Dissociated cells were washed with growth medium (DMEM/F12 containing 10% dextran-coated charcoal stripped FBS [DCC-FBS], 1% penicillin-streptomycin, 2 mM L-glutamine, 1 nM E2 [Sigma-Aldrich] and 2 mg/ml insulin [Sigma-Aldrich]). Samples were passed through a 40 µm cell sieve. EnSCs were collected from the flowthrough, while epithelial clumps remained in the sieve and were collected by backwashing into a 50 ml Falcon tube. Samples were resuspended in growth medium and centrifuged at 400× *g* for 5 min. EnSC pellets were resuspended in 10 ml growth medium and plated in tissue culture flasks. To isolate EnSC from other (non-adherent) cells collected in the flowthrough, medium was refreshed after 24 hr. Thereafter, medium was refreshed every 48 hr. Sub-confluent monolayers were passaged using 0.25% Trypsin-EDTA and split at a ratio of 1:3.

## Endometrial gland-like organoid culture

Endometrial gland-like organoids were established as described previously (*Turco et al., 2017*), with adaptations. Freshly isolated endometrial gland fragments were resuspended in 500 µl phenol red-free DMEM/F12 medium in a microcentrifuge tube and centrifuged at 600× *g* for 5 min. The medium was aspirated and ice-cold, growth factor-reduced Matrigel (Corning Life Sciences B.V., Amsterdam, Netherlands) was added at a ratio of 1:20 (cell pellet: Matrigel). Samples mixed in Matrigel were kept on ice until plating at which point the suspension was aliquoted in 20 µl volumes to a 48-well plate, one drop per well, and allowed to cure for 15 min. Expansion medium supplemented with E2 (*Supplementary file 1*: Table 1; *Turco et al., 2017*) was then added and samples cultured for up to 7 days. For passaging, Matrigel droplets containing gland-like organoids were collected into microcentrifuge tubes and centrifuged at 600× *g* for 6 min at 4 °C. Samples were resuspended in ice-cold, phenol red-free DMEM/F12 and subjected to manual pipetting to disrupt the organoids. Suspensions were centrifuged again, resuspended in ice-cold additive-free DMEM/F12, and then subjected to further manual pipetting. Suspensions were centrifuged again and either resuspended in Matrigel and plated as described above for continued expansion or used to establish assembloid cultures.

## Establishment of assembloid cultures

At passage 2, EnSC and gland-like organoid pellets were mixed at a ratio of 1:1 (v/v) and ice-cold PureCol EZ Gel (Sigma-Aldrich) added at a ratio of 1:20 (cell pellet: hydrogel). Samples were kept on ice until plating. The suspension was aliquoted in 20 µl volumes using ice-cold pipette tips into a 48-well plate, one droplet per well, and allowed to cure in the cell culture incubator for 45 min. Expansion medium supplemented with 10 nM E2 was overlaid and the medium was refreshed every 48 hr. For decidualization experiments, assembloid cultures were grown in expansion medium supplemented with E2 for 4 days to allow for growth and expansion. Assembloids were then either harvested or decidualized using different media as tabulated in *Supplementary file 1*: Table 1 for a further 4 days. Again, the medium was refreshed every 48 hr and spent medium stored for further analysis. For tyrosine kinase inhibition, MDM was supplemented with 250 nM dasatinib (Cell Signaling Technology, Leiden, NL).

## Fluorescence microscopy

For fluorescent microscopy, assembloids were removed from culture wells and transferred into tubes for fixation. Samples were washed in PBS and fixed in 10% neutral buffered formalin in the tube for 15 min, then washed three times with PBS, and stored for use. Samples were dehydrated in increasing concentrations of ethanol (70% then 90% for 1 hr each, followed by 100% for 90 min), then incubated in xylene for 1 hr. After paraffin wax embedding, 5 μm sections were cut and mounted, then incubated overnight at 60 °C. Slides were then stored at 4 °C until further processing. De-paraffinization and rehydration were performed through xylene, 100% isopropanol, 70% isopropanol, and distilled water incubations. Following antigen retrieval, permeabilization was performed where appropriate by incubation with 0.1% Triton X-100 for 30 min. Slides were then washed, blocked, and incubated in primary antibodies overnight at 4 °C. Antibody details are presented in *Supplementary file 1*: Table 3. After washing three times, slides were incubated with secondary antibodies for 2 hr, then washed as before and mounted in ProLong Gold Antifade Reagent with DAPI (Cell Signaling Technology). Slides were visualized using the EVOS Auto system, with imaging parameters maintained throughout image acquisition. Images were merged in ImageJ and any adjustments to brightness or contrast were applied equally within comparisons.

## Real-time quantitative polymerase chain reaction

After removal of spent medium, gland-like organoid cultures were washed in PBS and harvested in 200 μl Cell Recovery Solution (Corning). Gel droplets were transferred to nuclease-free microcentrifuge tubes and placed at 4 °C for 30 min. Samples were then washed in PBS, centrifuged at 600× $g$ for 6 min twice, and snap frozen as cell pellets. Assembloid cultures were washed with PBS and then recovered by directly scraping the samples into nuclease-free microcentrifuge tubes. Samples were centrifuged at 600× $g$ for 6 min. The cellular pellet was resuspended in 500 μl of 500 μg/ml collagenase I diluted in additive-free DMEM/F12 and incubated at 37 °C for 10 min with regular manual shaking. Samples were washed twice in PBS, with centrifugation at 600× $g$ for 6 min, then cell pellets were snap frozen. RNA extraction was performed using the RNeasy Micro Kit (QIAGEN, Manchester, UK) according to the manufacturer's instructions. RNA concentration and purity were determined using a NanoDrop ND-1000. All RNA samples were stored at –80 °C until use. Reverse transcription was performed using the QuantiTect Reverse Transcription (RT) Kit according to the manufacturer's protocol (QIAGEN). Input RNA was determined by the sample with lowest concentration within each experiment. Genes of interest were amplified using PrecisionPlus SYBR Green Mastermix (PrimerDesign, Southampton, UK). Amplification was performed in 10 μl reactions containing 5 μl PrecisionPlus 2× master mix, 300 nM each of forward and reverse primer, nuclease-free water, and 1 μl of cDNA or water control. Amplification was performed for 40 cycles on an Applied Biosystems QuantStudio 5 Real-Time PCR System (qPCR). Data were analysed using the Pffafl method (*Pfaffl, 2001*) and *L19* was used as a reference gene. Primer sequences were as follows: *L19* forward: 5′-GCG GAA GGG TAC AGC CAA T-3′, *L19* reverse: 5′-GCA GCC GGC GCA AA-3′, *PAEP* forward: 5′-GAG CAT GAT GTG CCA GTA CC-3′, *PAEP* reverse: 5′-CCT GAA AGC CCT GAT GAA TCC-3′, *SPP1* forward: 5′-TGC AGC CTT CTC AGC CAA A-3′, *SPP1* reverse: 5′-GGA GGC AAA AGC AAA TCA CTG-3′.

## Enzyme-linked immunosorbent assay

Spent medium was collected every two days during a 4- day decidual time course, with or without dasatinib treatment. Duoset solid-phase sandwich enzyme-linked immunosorbent assay (ELISA) kits (Bio-Techne, Abingdon, UK) were used for the detection of PRL (DY682), TIMP3 (DY973), IL-8 (DY208), IL-15 (DY247), CXCL14 (DY866), and HCG (DY9034). Assays were performed according to the manufacturer's instructions. Absorbance at 450 nm was measured on a PheraStar microplate reader (BMG LABTECH Ltd, Aylesbury, UK), with background subtraction from absorbance measured at 540 nm. Protein concentration was obtained using a four-parameter logistic regression analysis and interpolation from the curve. As medium was collected at different timepoints in a time-course culture, secreted levels were not normalized to total cell or protein contents.

## Single-cell capture, library preparation, and sequencing

Assembloids were dissociated to single cells by incubation of gel droplets with 0.5 mg/ml collagenase I for 10 min in a 37 °C water bath for 10 min with regular vigorous shaking. Samples were washed with

additive-free DMEM/F12 phenol-free medium and incubated with 5× TrypLE Select diluted in additive-free DMEM/F12 phenol-free medium for 5 min in a 37 °C water bath. Cell clumps were disrupted by manual pipetting, then suspended in 0.1% bovine serum albumin (BSA) in PBS and passed through a 35 μm cell sieve. Droplet generation was performed using a Nadia Instrument (Dolomite Bio, Cambridge, UK) according to the manufacturer's guidelines and using reagents as described by *Macosko et al., 2015* and the scRNAseq v1.8 protocol (Dolomite Bio). Pooled beads were processed as described previously (*Lucas et al., 2020*) and sequenced using a NextSeq 500 with high-output 75-cycle cartridge (Illumina, Cambridge, UK) by the University of Warwick Genomics Facility.

## Bioinformatics analysis

Initial single-cell RNAseq data processing was performed using Drop-Seq_tools-2.3.0 (DropseqAlignmentCookbook_v2Sept2018, http://mccarrolllab.com/dropseq) and as described previously (*Lucas et al., 2020*). To select high-quality data for analysis, cells were included when at least 200 genes were detected, while genes were included if they were detected in at least three cells. Cells with more than 5000 genes were excluded from the analysis as were cells with more than 5% mitochondrial gene transcripts to minimize doublets and low quality (broken or damaged) cells, respectively. The Seurat v3 standard workflow (*Stuart et al., 2019*) was used to integrate datasets from biological replicates. Clustering and nearest-neighbour analysis was performed on the full integrated dataset using principal components 1:15 and a resolution of 0.6. The 'subset' function was applied for interrogation of specific experimental conditions and timepoints. Gene Ontology (GO) analysis was performed on differentially expressed genes from specified 'FindMarkers' comparisons in Seurat v3 using the Gene Ontology Consortium database (*Ashburner, 2000*; *THE GENE ONTOLOGY, 2019*; *THE GENE ONTOLOGY, 2019*; *Mi et al., 2013*). Dot plots of significantly enriched GO terms (FDR-adjusted $p < 0.05$) were generated in RStudio (version 1.2.5042). CellPhoneDB was used to predict enriched receptor-ligand interactions between subpopulations in decidualizing assembloids (*Efremova et al., 2020*; *Vento-Tormo et al., 2018*). Significance was set at $p < 0.05$. Annotated tyrosine kinase interactions were curated manually.

## Co-culture of human blastocyst and endometrial assembloids

Prior to co-culture, decidualized assembloids were washed in PBS and medium was replaced with embryo medium (*Supplementary file 1*: Table 1). Assembloids were lightly punctured with a needle to create a small pocket, to enable one re-expanded day 5 human blastocyst to be co-cultured per assembloid. The plate was transferred to a pre-warmed and gassed (humidified 5% $CO_2$ in air) environment chamber placed on an automated X-Y stage (EVOS FL Auto Imaging System with onstage incubator) for time-lapse imaging. Brightfield images were captured every 60 min over 72 hr. Captured images were converted into videos using ImageJ.

For fixation, assembloid co-cultures were removed from culture wells and transferred into tubes. Samples were washed in PBS and fixed in 10% neutral buffered formalin in the tube for 15 min, then washed three times with PBS. Assembloids were permeabilized for 30 min in PBS containing 0.3% Triton X-100 and 0.1 M glycine for 30 min at room temperature. Samples were incubated overnight at 4 °C in primary antibodies diluted in PBS containing 10% FBS, 2% BSA, and 0.1% Tween-20. Samples were then washed in PBS (0.1% Tween-20) and incubated for 2 hr at room temperature protected from light in fluorescently conjugated Alexa Fluor secondary antibodies 1:500 (ThermoFisher Scientific) and DAPI (D3571, ThermoFisher Scientific, dilution 1/500), diluted in PBS containing 10% FBS, 2% BSA, and 0.1% Tween-20. Samples were imaged on a Leica SP8 confocal microscope using a ×25 water objective, with a 0.6 μm z-step and 2× line averaging.

## Statistical analysis

Data were analysed using GraphPad Prism. Pairwise comparison of non-parametric data was performed using Mann–Whitney U test. For paired, non-parametric significance testing between multiple groups, the Friedman test, and Dunn's multiple comparisons *post hoc* test were performed. Only values of $p < 0.05$ were considered statistically significant.

## Acknowledgements

We are grateful to the women and couples who participated in this research. We are indebted to Dr. Siobhan Quenby and all the staff in the Centre for Reproductive Medicine and Biomedical

Research Unit, University Hospitals Coventry and Warwickshire National Health Service Trust, for facilitating sample collection. This work was supported by a Wellcome Trust Investigator Award to JJB (212233/Z/18/Z). TMR was supported by the MRC Doctoral Training Partnership (MR/N014294/1) and a fellowship from Warwick-Wellcome Trust Translational Partnership initiative.

## Additional information

### Funding

| Funder | Grant reference number | Author |
|---|---|---|
| Wellcome Trust | 212233/Z/18/Z | Jan Joris Brosens |
| MRC Doctoral Training Partnership | MR/N014294/1 | Thomas M Rawlings |
| Warwick-Wellcome Trust Translational Partnership | | Thomas M Rawlings |

The funders had no role in study design, data collection and interpretation, or the decision to submit the work for publication.

### Author contributions

Thomas M Rawlings, Formal analysis, Investigation, Methodology, Project administration, Visualization, Writing – original draft, Writing – review and editing; Komal Makwana, Formal analysis, Investigation, Writing – review and editing; Deborah M Taylor, Joshua Odendaal, Amelia Hawkes, Geraldine M Hartshorne, Resources, Writing – review and editing; Matteo A Molè, Katherine J Fishwick, Maria Tryfonos, Investigation, Writing – review and editing; Magdalena Zernicka-Goetz, Supervision, Writing – review and editing; Jan J Brosens, Conceptualization, Formal analysis, Funding acquisition, Project administration, Resources, Supervision, Visualization, Writing – original draft, Writing – review and editing; Emma S Lucas, Formal analysis, Investigation, Methodology, Project administration, Supervision, Visualization, Writing – original draft, Writing – review and editing

### Author ORCIDs

Magdalena Zernicka-Goetz  http://orcid.org/0000-0002-7004-2471
Jan J Brosens  http://orcid.org/0000-0003-0116-9329
Emma S Lucas  http://orcid.org/0000-0002-8571-8921

### Ethics

Human subjects: Endometrial biopsies were obtained from women attending the Implantation Research Clinic, University Hospitals Coventry and Warwickshire National Health Service Trust. Written informed consent was obtained in accordance with the Declaration of Helsinki 2000. The study was approved by the NHS National Research Ethics Committee of Hammersmith and Queen Charlotte's Hospital NHS Trust (1997/5065) and Tommy's Reproductive Health Biobank (Project TSR19-002E, REC Reference: 18/WA/0356). The use of vitrified human blastocysts was carried out under a Human Fertilisation and Embryology Authority research licence (HFEA: R0155) with local National Health Service Research Ethics Committee approval (04/Q2802/26). Spare blastocysts were donated to research following informed consent by couples who had completed their fertility treatment at the Centre for Reproductive Medicine, University Hospitals Coventry and Warwickshire National Health Service Trust.

### Decision letter and Author response

Decision letter https://doi.org/10.7554/eLife.69603.sa1
Author response https://doi.org/10.7554/eLife.69603.sa2

## Additional files

### Supplementary files

• Supplementary file 1. Supplementary tables 1-3.
 Table 1. Culture media composition. Table 2. Patient demographics for endometrial samples. Table

3. Antibody details for immunofluorescence labelling.

• Transparent reporting form

## Data availability

Single cell RNAseq data presented in this paper are openly available as a Gene Expression Omnibus DataSet (https://www.ncbi.nlm.nih.gov/gds) under accession number GSE168405. Other source data are presented in the Source Data tables as indicated in the corresponding Figure legends.

The following dataset was generated:

| Author(s) | Year | Dataset title | Dataset URL | Database and Identifier |
|---|---|---|---|---|
| Brosens JJ, Lucas ES, Rawlings TM | 2021 | Single-cell RNA Sequencing of Endometrial Assembloid Cultures | https://www.ncbi.nlm.nih.gov/geo/query/acc.cgi?acc=GSE168405 | NCBI Gene Expression Omnibus, GSE168405 |

The following previously published datasets were used:

| Author(s) | Year | Dataset title | Dataset URL | Database and Identifier |
|---|---|---|---|---|
| Talbi S, Hamilton AE, Kc Vo, Tulac S, Overgaard MT, Dosiou C, Le Shay N, Le Shay N, Kimpson R, Lessey BA, Nayak NR, Giudice LC | 2006 | Molecular phenotyping of human endometrium | https://www.ncbi.nlm.nih.gov/geo/query/acc.cgi?acc=GSE4888 | NCBI Gene Expression Omnibus, GSE4888 |

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
