## [Decision Letter]

**Acceptance summary:**

This paper elegantly combines single cell transcriptomic and novel three-dimensional culture models of the human endometrium to reveal how cell senescence impacts the ability of the uterus to prepare and support embryo implantation. This work provides important insights into the etiology of pregnancy failure in women and may accelerate the discovery of new treatments to improve reproductive health.

**Decision letter after peer review:**

Thank you for submitting your article "Modelling the impact of decidual senescence on embryo implantation in human endometrial assembloids" for consideration by *eLife*. Your article has been reviewed by 4 peer reviewers, including Thomas E Spencer as the Reviewing Editor and Reviewer #1, and the evaluation has been overseen by Jonathan Cooper as the Senior Editor. The following individuals involved in review of your submission have agreed to reveal their identity: Gunter Wagner (Reviewer #2); Hugo Vankelecom (Reviewer #3).

Essential revisions:

1) The story is not always easy to follow if not completely submerged in this field of endometrial receptivity, decidualization, importance of senescence, of NK cells, etc. Authors should at certain points be more focused and should less diverge from the subject (certainly in the Discussion) which would better retain the attention of the reader to the central message of the study. Also, the Abstract could be simplified in this regard. Moreover, the paper will highly benefit from a graphical abstract of summarizing figure, since messages conveyed are not easy for the broader reader.

2) The epithelial organoid cultures are referred to as glandular organoids throughout the manuscript; however, the cultures may contain a mixed population of glandular and luminal epithelial cells. It would be beneficial to clarify the proportion of glandular and luminal epithelial cells within the culture. This can be done using established markers of endometrial glandular epithelial cells such as FOXA2.

3) The authors should consider testing their minimal differentiation medium to determine whether it's sufficient to induce a stromal cell response, rather than only testing the secretory response of the endometrial epithelium. If these tests have been done, the authors should consider mentioning them at the place in the text where the epithelial secretion response is discussed.

*Reviewer #1 (Recommendations for the authors):*

1. As the manuscript introduces a new co-culture model of endometrial stroma and epithelial cells, the authors must describe the relative proportion of cells used to establish co-culture. Specifically, a 1:1 (v/v) ratio of different cell types is ill-defined and will make replication difficult as pellet size can depend on the amount of organoid disruption. Please clarify to enhance reproducibility.

2. The epithelial organoid cultures are referred to as glandular organoids throughout the manuscript; however, the cultures may contain a mixed population of glandular and luminal epithelial cells. It would be beneficial to clarify the proportion of glandular and luminal epithelial cells within the culture. This can be done using established markers of endometrial glandular epithelial cells such as FOXA2.

3. Although the authors have introduced a new model that likely will advance our understanding of implantation and pregnancy loss in the future, it is unclear if it provides an advancement in the study of uterine stromal cell decidualization. An unaddressed question from this study is the impact of epithelial cells on stromal cell decidualization. In the future, it will be necessary to directly compare this model with 3D stromal cell culture in the absence of epithelial cells.

4. In Figure 6, the authors should provide images in the z-plane to show definitive attachment. Likewise, it is advisable to define firm attachment in line 349. As this is only observed as an n=1, the location of the embryo may be a fixation artifact. Additionally, it would not be expected for the trophectoderm to attach to glandular organoids on the side of the basal lamina. Thus, the authors should address the polarity of the organoids should be explored.

5. Line 259 refers to EpC5 when it should be EpS5.

*Reviewer #2 (Recommendations for the authors):*

P 7: I would be interested in your thinking about the role of NAC in the base medium. There is a role for NOX4 and ROS in decidualization, so should NAC not be inhibiting decidualization?

Line 157: the role of the level of stress is not really addressed in this paper as I see it, i.e. there is no comparison between more or less replication. May be this statement is not helpful here.

Also it seems not clear how your experimental model deals with differences between luteal phase changes and those caused by the embryo. Do you expect cAMP signaling be part of the spontaneous decidualization in the cycle or dependent on embryo attachment?

Line 237: PDGFRA and -B are receptors not growth factors.

Line 276: EpC4 should be EpS4.

281 to 282: isn't that always the case?

Random thought: Given what you write about spread of senescence to neighboring cells, I am wondering whether menstruation results from the withdrawal of DSC directed recruitment of uNK???

*Reviewer #3 (Recommendations for the authors):*

1) The story is not always easy to follow if not completely submerged in this field of endometrial receptivity, decidualization, importance of senescence, of NK cells, etc. Authors should at certain points be more focused and should less diverge from the subject (certainly in the Discussion) which would better retain the attention of the reader to the central message of the study. Also, the Abstract could be simplified in this regard. Moreover, the paper will highly benefit from a graphical abstract of summarizing figure, since messages conveyed are not easy for the broader reader.

2) Although the term assembloids has mostly been used for constructs combining organoids from different tissues or multiple cell types, it sounds okay and can be used here.

3) The use of a single, very broad tyrosine kinase (TK) inhibitor seems a bit crude as perturbation approach. Not all TKs act in similar processes. Do the authors have an idea about more specific TK pathways that may be involved, starting from their data? It could be interesting to just touch on one of them to zoom in on more specific molecular mechanisms. Or will authors elaborate on this in follow-up studies? Moreover, the effect of dasatinib could also be briefly confirmed with another broad TK inhibitor (although experiments with a more specific one would be rather preferred to fine-tune the focus). In lines 424-426, authors claim a role for the "level of endogenous cellular stress …" from the impact of dasatinib; however, TK signaling does not only underlie celluar stress. Please explain this better.

4) Authors describe the epithelial compartment in the organoids/assembloids as glandular. Do they have indications from their scRNA-seq data that may point to presence of some luminal epithelium? As far as the reviewer remembers, other scRNA-seq studies may have shown this (Fitzgerald et al., PNAS 2019; Cohrane group; Turco group in BioRxiv). Please discuss.

5) The study should briefly discuss similarities and differences with a recent paper of the Kessler group (Cell Rep 2021) also having designed an epithelial-stromal co-culture model.

Specific comments on the different parts:

Impact statement:

– " … reveals novel mechanisms of reproductive failure …" should be somewhat toned down since not extensively supported yet, e.g. to "… may lead to new insights into …" or " … will help to unravel/decipher …".

Abstract:

– line 37-38: not understandable for the broader reader.

– line 40: "… mesenchymal-epithelial transition, processes involved in endometrial breakdown and regeneration": is this sufficiently accepted among endometrium researchers to include it in the Abstract? I would suggest to remove this here.

– Line 46-47: is rather hypothetical for an abstract.

Introduction:

– Line 67: "phenotypic decidual cells"? Please explain or rephrase.

– Lines 107-109: "… demonstrate that different pathological states can be recapitulated …": this conclusion sounds too far, since authors use only one approach (i.e. the broad TK inhibitor), which cannot be immediately extrapolated to multiple ("different") pathological states. This generality is indeed not supported yet by experiments; some more proof-of-principle experiments would then be needed (e.g. even Crispr/Cas could be considered). This conclusion is repeated in lines 361-362 and should also there be moderated.

Results:

– Line 111, 153: "simple": why is this adjective used? As compared to?

– Line 113: "progenitor cells": it is not known whether organoids are formed by "progenitors" (neither is there conclusive evidence that they exist); this should be removed.

– Line 122: also Boretto et al. 2017 should be mentioned here.

– Line 127: how was the 1:1 ratio defined or controlled since organoids were not dissociated into single cells (see Methods line 539)? This is important technical information for a new model.

– Line 137: "… mimic luteal phase endometrium": please, show some stainings of primary tissue to support this.

– Line 162-1645: authors compare day 4 assembloids (undifferentiated), thus being in a proliferative phase, with assembloids in decidualized phase (meaning 4 days longer in culture in non-proliferative conditions). I guess the authors took this into account for DEG/GO analysis, by extracting the impact of cell cycle genes? Please explain.

– Lines 172-173: "actively dividing EpC" and "EpC with marker genes of the E2-responsive proliferative endometrium": both are proliferative, so what is their exact difference then? What are the in vivo counterparts of both subclusters? Same question applies to the stromal subclusters SS1 and SS2 (lines 206-207).

– Line 180: canonical endometrial receptivity genes: please indicate them in the figure, or mention a few here in the text.

– Lines 193 …: regarding the transitional population (TP): what may be the in vivo correlate? What could be the in vivo function of the TP? The TP appears not to be found in the in vivo endometrium (see Figure 3C). Line 257: there is a lack of crosstalk of TP with other cells, so is the TP an in vitro population not present in vivo? Should be discussed.

– Line 218: "… novel pre-decidual genes were also identified" … moderate to "novel candidate pre-decidual genes …".

– Lines 247…: "… transition between cellular states is predicated …". This could be further supported by applying pseudotime analysis (using Monocle, RNAvelocity, …). Did the authors consider or perform this?

– Line 299: dasatinib needs somewhat more introduction/explanation here. How does it work? What does it exactly do?

– Line 315: TP develop by MET: what are the predicted source cells then? No relevant information available from pseudotime analysis?

– Line 328-331: this conclusion is not clear from the data; please rephrase and/or include in graphical abstract.

– Line 342: please immediately specify that exactly one embryo is added per assembloid drop/well.

– Line 358: all embryo cocultures were found to secrete hCG; thus the 'receptive' co-cultures (without dasatinib) do not show more features of correct interaction ('pregnancy') than the dasatinib-treated ones? Please discuss.

– Some typo's: line 259: EpS5; line 276: EpS4.

Discussion:

– Lines 364-373: This part is not really to the point, and could be removed to better retain the focus.

– Line 423: "… MET drives re-epithelialization …". how sure is this in the field? Has this recently been unequivocally underpinned? If not, please moderate.

Methods:

– Line 526-527: if I understand well, authors use 95% Matrigel/5% cells in medium (1:20), which is high and completely different from other studies using lower % of Matrigel; please confirm or explain.

– ELISAs: how were secretions in the medium normalized to the number of organoids/cells present in the assembloid drop? Were they normalized to extracted cellular protein b-actin? Or in other words, how was the number of organoids/cells in the structures standardized among wells/plates/independent experiments for secreted protein measurements?

Figures:

– Figure 2: PAEP and SPP1 in ExM/E2/cAMP/MPA are indicated as non-significantly different; is this true or correct?

– Figure 3 C and E: in vivo: please add reference of these data in the legend.

– Figure 6D: right figure is not explained.

– Figure 6: "… attached by proliferating polar trophectoderm": is "proliferating" supported by experimental evidence? Please show then.

*Reviewer #4 (Recommendations for the authors):*

Suggestions for improved or additional experiments

– The authors should consider testing their minimal differentiation medium to determine whether it's sufficient to induce a stromal cell response, rather than only testing the secretory response of the endometrial epithelium. If these tests have been done, the authors should consider mentioning them at the place in the text where the epithelial secretion response is discussed.

– Line 166 and 167 mentions the dotted line in figure 3B. Please add an explanation for the dotted line circles.

– Line 172-173: Please provide more clarity on the difference between actively dividing epithelial cells and epithelial cells expressing markers of proliferating endometrium.

– Line 218: Were these novel pre-decidual genes verified in human pre-decidual samples? Can the authors confirm that they are not artifacts of their culture system?

– Please clarify how ligands and receptors were chosen to be entered into the CellPhoneDB program.

– Line 260: Please clarify the CellPhoneDB data. It appears that only the RNA expression data is used for this experiment, so can it be assumed that the two proteins are both interacting, even if they are both expressed? The CellPhoneDB data should be validated.

– Line 316: Why does the fact that dasatinib upregulates mesenchymal genes and decidual factors in transitional cells show that MET causes the transitional cells to emerge during decidualization? More evidence is needed for this claim.

– Line 361: Please clarify which different pathological endometrial states were recapitulated.

– Is there data demonstrating that implantation failure is associated with insufficient senescence?

– The "assembloids" are new but there are renditions of the 3D models using epithelial cells and stromal cells that are published and could be cited for comparison purposes. Given the different way the stromal cells were cocultured (in hydrogel) in the assembloid, the morphology is unique and the transcriptomics are new.

– Single cell RNA-seq of organoids have been performed by Spencer's group although they did not have the cocultured stromal cells. In that sense, the scRNA-seq of epithelial/stromal assembloids have revealed new data and it would be interesting to determine how epithelial transcriptomics changed in the presence of stromal cells.

[Editors' note: further revisions were suggested prior to acceptance, as described below.]

Thank you for submitting your article "Modelling the impact of decidual senescence on embryo implantation in human endometrial assembloids" for consideration by *eLife*. Your article has been reviewed by 4 peer reviewers, including Thomas E Spencer as the Reviewing Editor and Reviewer #1, and the evaluation has been overseen by Jonathan Cooper as the Senior Editor. The following individuals involved in review of your submission have agreed to reveal their identity: Gunter Wagner (Reviewer #2); Hugo Vankelecom (Reviewer #3); Ji-Yong Julie Kim (Reviewer #4).

Essential revisions:

Please address the suggestions from the reviewers and submit a final version of the manuscript.

*Reviewer #1 (Recommendations for the authors):*

The authors have satisfactorily addressed the majority of the comments and suggestions from the initial review. They are encouraged to submit a final manuscript that incorporates the remaining suggestions of the reviewers.

*Reviewer #2 (Recommendations for the authors):*

Thank you for the revision of the paper, and congratulations to that exciting paper. I am happy with the revision, with a few minor suggestions:

Line 95: please clarify what you mean by "turnover" of senescent cells. Do you mean removal? Since turnover seems to mean elimination and replacement. How does turnover recruit decidual precursor cells? Or do you mean secretions from senescent cells recruit mesenchymal precursor cells?

The Diniz-da-Costa reference is incomplete.

Line 380ff: not sure whether it is appropriate to say that SC analysis, which is a descriptive tool, gives evidence for causality (requires divergence…).

*Reviewer #3 (Recommendations for the authors):*

The authors responded appropriately to the suggestions and remarks, and very well clarified the points of confusion. I have no further comments.

*Reviewer #4 (Recommendations for the authors):*

The authors have addressed most of Reviewer 4's concerns. However, it remains important that other epithelial and stromal organoid systems are cited in this study, as also suggested by Reviewer 3.

Overall, this is a lovely study.

---

## [Author Response]

Essential revisions:1) The story is not always easy to follow if not completely submerged in this field of endometrial receptivity, decidualization, importance of senescence, of NK cells, etc. Authors should at certain points be more focused and should less diverge from the subject (certainly in the Discussion) which would better retain the attention of the reader to the central message of the study. Also, the Abstract could be simplified in this regard. Moreover, the paper will highly benefit from a graphical abstract of summarizing figure, since messages conveyed are not easy for the broader reader.

The current implantation paradigm in humans is informed by animal models, foremost by studies in mice. Consequently, the implantation process is often reduced and equated to breaching of the luminal epithelium by an embryo. In mice, a polytocous rodent, the barrier function of the luminal epithelial plays a critical role in synchronising implantation of multiple embryos, which can transiently arrest development (diapause) in the uterine cavity while awaiting a maternal nidation signal (estrogen). Implantation in humans, on the other hand, normally involves a single embryo, often harbouring complex chromosomal errors and lacking the ability to transiently arrest development while awaiting a maternal implantation signal (which does not appear to exist). Hence, whether the barrier function of the luminal endometrium epithelium in mice is maintained or degraded in humans is questionable and, by extension, so is the prevailing implantation paradigm. More importantly, the major hurdle at implantation in humans, and other menstruating primates, is not synchronised implantation of multiple blastocysts, but transformation of cycling endometrium into the decidua of pregnancy robust enough to accommodate a deeply invading placenta throughout gestation. This process inevitably requires intense tissue remodelling and the underlying mechanisms that control endometrial fate decision at implantation are explored in this study. Hence, we make no apologies for describing the major cellular players in this process. In fact, all the players involved in endometrial remodelling (acute senescent cells, differentiated cells, innate immune cells, and chronic senescent cells) will be instantaneously recognisable to anyone working in the field of tissue remodelling, whether in the context of development, wound healing, ageing or even cancer. Nevertheless, we appreciate this request for more clarity. Hence, we have produced a Graphical Abstract, simplified the Abstract, and amended the Discussion.

2) The epithelial organoid cultures are referred to as glandular organoids throughout the manuscript; however, the cultures may contain a mixed population of glandular and luminal epithelial cells. It would be beneficial to clarify the proportion of glandular and luminal epithelial cells within the culture. This can be done using established markers of endometrial glandular epithelial cells such as FOXA2.

The use of the word ‘glandular’ was intended to indicate the in vitro characteristics of the tubular, gland-like organisation of epithelial cells in assembloids, rather than the identity of the progenitor cells, but we see how this can be misconstrued. We note that the Turco group used the same terminology in their original report (PMID: 28394884), as well as describing the establishment of the organoid cultures from ‘retained glandular elements … backwashed from the sieve membranes’. Nevertheless, we have changed the term ‘glandular’ organoids to ‘gland-like’ organoids to avoid confusion.

We note the reviewer’s suggestion to explore the proportions of glandular and luminal cells and attempted to do so by examining the expression of putative maker genes in the different populations of undifferentiated and decidualized assembloids. However, constitutive expression of commonly described markers is not apparent either in whole tissue (see GEO Dataset GDS2053: Endometrium through the Menstrual Cycle, and PMID: 16306079) nor in our assembloid cultures (Author response image 1), and markers described as being specific to luminal epithelium are expressed and temporally regulated at RNA level in laser capture micro-dissected glands in vivo (Author response image 1).

**Author response image 1. sa2fig1:** (A) Expression of FOXA2, a glandular epithelial marker (green font), and putative luminal epithelial marker genes (black font) in assembloid subpopulations. Dot size indicates the proportion of cells expressing the marker, while colour indicates the level of expression. Note that none of the cell populations express a compelling luminal epithelial marker gene signature. (B) Expression of the same markers in laser-capture micro-dissected endometrial glands in vivo, obtained during the early- and mid-luteal phase (LH^+^5 and LH^+^8, respectively).

Furthermore, putative protein markers identified in tissue sections (e.g. PMID: 28394884 and Garcia-Alonso, bioRxiv https://doi.org/10.1101/2021.01.02.425073) are not well-represented at the RNA level, perhaps due to the dropout phenomenon observed in single-cell datasets (see PMID: 32127540). Further, in the Garcia-Alonso preprint article, label transfer from in vivo single-cell data onto the gland-like organoid dataset reveals very few luminal epithelial cells to our eyes, and expression of luminal epithelial markers (e.g. LGR5, *SOX9*,WNT7A^high^) appears lost upon hormonal differentiation of organoids. It is also notable that the overlap in epithelial gene expression between gland fragments and organoids reported by Turco et al., (PMID: 28394884) was fairly low (30-37%), suggesting that cells do undergo a phenotypic change in vitro, although the full gene lists are not available alongside that paper.

3) The authors should consider testing their minimal differentiation medium to determine whether it's sufficient to induce a stromal cell response, rather than only testing the secretory response of the endometrial epithelium. If these tests have been done, the authors should consider mentioning them at the place in the text where the epithelial secretion response is discussed.

We have now tested the decidualization response of primary endometrial stromal cells in standard 2D cultures to the minimal differentiation medium supplemented with E2, cAMP and MPA (MDM+E2+cAMP+MPA) in comparison to our standard differentiation medium for 2D cultures (2% DCC-DMEM+cAMP+MPA). Further, the impact of assembloid expansion medium supplemented with E2 (ExM+E2) during the growth phase was compared to that of our standard growth medium (10% DCC-DMEM) for primary cultures. Secreted levels of IL-8 (released by pre-decidual cells without corresponding induction at mRNA level) and CXCL14 and TIMP3 (induced upon decidualization) were monitored in two biologically independent cultures. Unfortunately, access to endometrial biopsies is currently very limited because of the impact of the pandemic on our research clinic. As shown in Author response image 2, kinetics of decidual secretions were identical under all treatment conditions, characterised by a rapid but transient rise in IL-8 secretion on day 2 of decidualization and followed by accelerated rise in CXCL14 and TIMP3 secretion by day 4. Further, when subjected to the same differentiation medium (2% DCC-DMEM+cAMP+MPA), primary endometrial stromal culture grown first in ExM+E2 tended to secrete lower IL-8 levels on day 2 of decidualization but higher CXCL14 and TIMP3 levels by day 4 when compared to cultures first grown in 10% DCC-DMEM. These observations reinforce our assertion that the amplitude of initial pre-decidual stress response is determined during the preceding proliferative phase and that a blunted pre-decidual stress response accelerates decidualization and vice versa. We have reported similar observations previously (PMID: 31965050).

**Author response image 2. sa2fig2:** Comparison of secreted levels of IL-8, CXCL14 and TIMP3 in two independent primary endometrial stromal cells maintained in standard 2D cultures in response to standard growth medium (10% DCC-DMEM+E2) or assembloid expansion medium (ExM+E2) and standard differentiation medium (2% DCC-DMEM+cAMP+MPA) or assembloid differentiation medium (MDM+E2+cAMP+MPA), as indicated.

Reviewer #1 (Recommendations for the authors):1. As the manuscript introduces a new co-culture model of endometrial stroma and epithelial cells, the authors must describe the relative proportion of cells used to establish co-culture. Specifically, a 1:1 (v/v) ratio of different cell types is ill-defined and will make replication difficult as pellet size can depend on the amount of organoid disruption. Please clarify to enhance reproducibility.

Thank you for raising this issue. The stated 1:1 (v/v) ratio upon mixing of the different cell types equates to 5×10^4^ stromal cells and epithelial organoid fragments passaged at a ratio of 1:2 per assembloid culture. As mixing involved passaged organoid fragments, we cannot elaborate precisely on the number of individual epithelial cells. Nevertheless, this protocol is robust. The procedure has been clarified in the methods section.

2. The epithelial organoid cultures are referred to as glandular organoids throughout the manuscript; however, the cultures may contain a mixed population of glandular and luminal epithelial cells. It would be beneficial to clarify the proportion of glandular and luminal epithelial cells within the culture. This can be done using established markers of endometrial glandular epithelial cells such as FOXA2.

We refer the Reviewer to our previous response (Essential Revisions, point 2).

3. Although the authors have introduced a new model that likely will advance our understanding of implantation and pregnancy loss in the future, it is unclear if it provides an advancement in the study of uterine stromal cell decidualization. An unaddressed question from this study is the impact of epithelial cells on stromal cell decidualization. In the future, it will be necessary to directly compare this model with 3D stromal cell culture in the absence of epithelial cells.

Thank you – we appreciate this point. In fact, as part of a different project, we are assessing if stromal cells from control subjects can attenuate epithelial defects in gland-like organoids of women with recurrent missed miscarriage. As implied by the Reviewer, endometrial assembloids are indeed a powerful tool to dissect how defects in one cellular compartment influence differentiation response in the other compartment.

4. In Figure 6, the authors should provide images in the z-plane to show definitive attachment. Likewise, it is advisable to define firm attachment in line 349. As this is only observed as an n=1, the location of the embryo may be a fixation artifact. Additionally, it would not be expected for the trophectoderm to attach to glandular organoids on the side of the basal lamina. Thus, the authors should address the polarity of the organoids should be explored.

The time-lapse imaging shows decidualizing stromal cells extending toward the polar trophectoderm of the blastocyst and making contact by 46-50 hours in co-culture. Please see Figure 6—figure supplement 1, where stromal cell migration is demarcated by a dotted white line. Migrating cells continue to surround the embryo at this site and then draw the embryo toward the assembloid matrix. We are confident in the attachment of this embryo since it remained associated with the assembloid through transfer from the culture well into fixation and washing solutions, postage and staining processes while unattached embryos were lost during processing. However, we have removed the word ‘firm’ from the text to avoid potential overstatement. Future work will involve more extensive imaging and the use of additional labels to confirm implantation and outgrowth. Interestingly, endoglandular trophoblast invasion has been reported in human pregnancies (PMID: 13362122, PMID: 26493408), including at very early post-implantation. So, while we agree that initial embryo-endometrium interaction in vivo would not be expected to involve the basal lamina of glands, this may well occur shortly after implantation.

5. Line 259 refers to EpC5 when it should be EpS5.

Thank you. We have corrected this error.

Reviewer #2 (Recommendations for the authors):P 7: I would be interested in your thinking about the role of NAC in the base medium. There is a role for NOX4 and ROS in decidualization, so should NAC not be inhibiting decidualization?

We agree with the Reviewer that the mechanisms accounting for the beneficial action of NAC in our model warrant further exploration. However, the concentration of NAC used in the minimal differentiation medium is an order of magnitude lower than needed for ROS clearance (MDM contains 1.25 mM NAC, 10-15 mM NAC) is used elsewhere (PMID: 17343919, PMID: 19498006, PMID: 31885807, PMID: 33585475). Therefore, we speculate that low-dose NAC may not constrain decidualization-associated ROS production in assembloids, yet confer some protection against oxidative cell death and therefore beneficial upon differentiation. Additional roles for NAC include serving as a reserve of amino acid cysteine and maintaining redox homeostasis, which may also contribute to its beneficial effects in minimal differentiation medium (PMID: 18671159, PMID: 20602078). We have now stated in the text that NAC was used at low concentration.

Line 157: the role of the level of stress is not really addressed in this paper as I see it, i.e. there is no comparison between more or less replication. May be this statement is not helpful here.

In line with previously reported observations (PMID: 31965050), the dastinib experiments reinforces our assertion that the level of stress in pre-decidual cells (for example, measured by the secreted levels of IL-8), determines the kinetics and quality of the subsequent decidual response.

Also it seems not clear how your experimental model deals with differences between luteal phase changes and those caused by the embryo. Do you expect cAMP signaling be part of the spontaneous decidualization in the cycle or dependent on embryo attachment?

Endometrial cAMP levels rise sharply upon transition from the proliferative to secretory phase in non-conception cycles. We do not have information on whether embryonic cues stimulate cAMP/PKA signalling in endometrial stromal cells. In a different project, we exhaustively investigated if hCG increases cAMP levels in primary stromal cells but found no evidence for this, contrary to some claims in the literature.

Line 237: PDGFRA and -B are receptors not growth factors.

Thank you. We have amended the text.

Line 276: EpC4 should be EpS4.

Thank you. We have corrected this error.

281 to 282: isn't that always the case?

Apologies but we are unsure what the Reviewer is referring to.

Random thought: Given what you write about spread of senescence to neighboring cells, I am wondering whether menstruation results from the withdrawal of DSC directed recruitment of uNK???

Absolutely! We believe indeed that this is indeed the case. In the absence of an implanting embryo, and in response to falling progesterone (which disables the uNK-decidual cell partnership/cooperation), unopposed bystander senescence driven by mounting SASP is anticipated to cause sterile inflammation in the superficial layer, triggering a concatenation of events that ultimately results in menstrual breakdown. Across the menstrual cycle, expression of canonical senescence-associated genes generally rises sharply during the late-secretory phase. It is also plausible that transient senescence-associated inflammation in the superficial layer prior to menstruation ‘primes’ endometrial progenitors residing in the basal layer to promote tissue regeneration in the next cycle. The biological basis for this assertion is that transient SASP induces de-differentiation of cells to a more progenitor-like state and ‘lock-in’ of stem cells, thereby accelerating tissue regeneration upon resolution of SASP. Thus, it is conceivable, if not likely, that endometrial repair mechanisms become poised to be activated prior to menstruation. Notably, persistent, chronic SASP causes stem cell depletion and reduces regenerative capacity of tissues, a process widely believed to drive ageing and age-related disorders.

Reviewer #3 (Recommendations for the authors):1) The story is not always easy to follow if not completely submerged in this field of endometrial receptivity, decidualization, importance of senescence, of NK cells, etc. Authors should at certain points be more focused and should less diverge from the subject (certainly in the Discussion) which would better retain the attention of the reader to the central message of the study. Also, the Abstract could be simplified in this regard. Moreover, the paper will highly benefit from a graphical abstract of summarizing figure, since messages conveyed are not easy for the broader reader.

We refer the Reviewer to our previous response (Essential Revision 1).

2) Although the term assembloids has mostly been used for constructs combining organoids from different tissues or multiple cell types, it sounds okay and can be used here.

Thank you.

3) The use of a single, very broad tyrosine kinase (TK) inhibitor seems a bit crude as perturbation approach. Not all TKs act in similar processes. Do the authors have an idea about more specific TK pathways that may be involved, starting from their data? It could be interesting to just touch on one of them to zoom in on more specific molecular mechanisms. Or will authors elaborate on this in follow-up studies? Moreover, the effect of dasatinib could also be briefly confirmed with another broad TK inhibitor (although experiments with a more specific one would be rather preferred to fine-tune the focus). In lines 424-426, authors claim a role for the "level of endogenous cellular stress …" from the impact of dasatinib; however, TK signaling does not only underlie celluar stress. Please explain this better.

We agree that the broad action of dasatinib limits mechanistic understanding of senescent cell clearance in our assembloid cultures. Our data show enrichment of *SRC*, *ABL2* and *EPHB1* expression in the senescent stromal cells (SS5), suggesting that these are the key cell survival pathways that could be targeted by dasatinib. By contrast, the transitional population (TP) shows enriched expression of *EPHB2* and *EPHB4*, suggesting a different mode of action for dasatinib in these cells. The molecular mechanisms for senescence clearance using more specific inhibitors is currently under investigation.

**Author response image 3. sa2fig3:** 

4) Authors describe the epithelial compartment in the organoids/assembloids as glandular. Do they have indications from their scRNA-seq data that may point to presence of some luminal epithelium? As far as the reviewer remembers, other scRNA-seq studies may have shown this (Fitzgerald et al., PNAS 2019; Cohrane group; Turco group in BioRxiv). Please discuss.

We refer the Reviewer to our previous response (Essential Revisions, point 2).

5) The study should briefly discuss similarities and differences with a recent paper of the Kessler group (Cell Rep 2021) also having designed an epithelial-stromal co-culture model.

While there are parallels between the models, these authors used iPSC-derived fibroblasts to establish their 3D culture but were unable both in this and previous study (PMID: 32937244) to develop the model using primary stromal cells. The reasons for this are not entirely clear but may relate to the use of the basement membrane matrix, Matrigel, in their model. Indeed, clustering of their stromal lineage is very restricted to the periphery of the organoids, while our stromal cells form a matrix throughout the gel droplet. Thus, the approach taken by Cheung et al., i.e. using iPSC-derived stromal cells, has the advantage of being able to model mechanisms of lineage commitment. However, our model successfully incorporates primary endometrial stromal cells in a collagen matrix to model the functional layer of the endometrium. This offers a more physiological approach to study implantation processes in patient-specific assembloids, investigate endometrial dyshomeostasis, and evaluate therapeutic intervention. Additionally, our characterisation of a minimal differentiation medium further advances a more physiological model, enabling intrinsic cell-cell communications to regulate growth and differentiation.

Specific comments on the different parts:Impact statement:– " … reveals novel mechanisms of reproductive failure …" should be somewhat toned down since not extensively supported yet, e.g. to "… may lead to new insights into …" or " … will help to unravel/decipher …".

Thank you – done.

Abstract:– line 37-38: not understandable for the broader reader.

Thank you, amended.

– line 40: "… mesenchymal-epithelial transition, processes involved in endometrial breakdown and regeneration": is this sufficiently accepted among endometrium researchers to include it in the Abstract? I would suggest to remove this here.– Line 46-47: is rather hypothetical for an abstract.

Thank you. It is but we have nevertheless simplified the abstract.

Introduction:– Line 67: "phenotypic decidual cells"? Please explain or rephrase.

Thank you, done.

– Lines 107-109: "… demonstrate that different pathological states can be recapitulated …": this conclusion sounds too far, since authors use only one approach (i.e. the broad TK inhibitor), which cannot be immediately extrapolated to multiple ("different") pathological states. This generality is indeed not supported yet by experiments; some more proof-of-principle experiments would then be needed (e.g. even Crispr/Cas could be considered). This conclusion is repeated in lines 361-362 and should also there be moderated.

We are confident of these statements as the findings from this study complement and reinforce our previous observations using patient samples and primary cultures (and indeed ongoing studies). However, we appreciate the cautioning against overstated claims. Hence, we have rephrased this claim as ‘ …that aspects of different pathological states…’.

Results:– Line 111, 153: "simple": why is this adjective used? As compared to?

Thanks, we removed this inappropriate adjective throughout.

– Line 113: "progenitor cells": it is not known whether organoids are formed by "progenitors" (neither is there conclusive evidence that they exist); this should be removed.

Thanks, done.

– Line 122: also Boretto et al. 2017 should be mentioned here.

Thank you, we have included this reference.

– Line 127: how was the 1:1 ratio defined or controlled since organoids were not dissociated into single cells (see Methods line 539)? This is important technical information for a new model.

Thank you. As mentioned in response to Reviewer 1 (Comment 1), the 1:1 (v/v) ratio of the different cell type pellets equates to 5×10^4^ stromal cells and epithelial organoid fragments passaged at a ratio of 1:2 per assembloid culture. The reference to the ratio in line 123 has been removed, and the procedure has been clarified within the methods section.

– Line 137: "… mimic luteal phase endometrium": please, show some stainings of primary tissue to support this.

The spatiotemporal expression of these markers in secretory endometrium is well-described in the literature, e.g. PMID: 9022601, PMID: 9021377 (Laminin); PMID: 11591413, PMID: 29420252 (Osteopontin); PMID: 25695723, PMID: 22215622 (Glycodelin) and PMID: 22215622, PMID: 9620842 (PGR).

– Line 162-1645: authors compare day 4 assembloids (undifferentiated), thus being in a proliferative phase, with assembloids in decidualized phase (meaning 4 days longer in culture in non-proliferative conditions). I guess the authors took this into account for DEG/GO analysis, by extracting the impact of cell cycle genes? Please explain.

We appreciate the reviewer’s comment on our experimental design. As outlined in our response to a similar comment from Reviewer 2, our experimental design mimics the temporal transition from proliferative to secretory phase endometrium. Cell cycle phase was determined in our scRNAseq data according to the Seurat standard workflow. However, we made a conscious decision not to perform regression of cell cycle gene expression due to studying a model of differentiation wherein a switch in cell cycle state from proliferative (S, G2/M) to a predominantly non-dividing (G1/0) status is expected.

– Lines 172-173: "actively dividing EpC" and "EpC with marker genes of the E2-responsive proliferative endometrium": both are proliferative, so what is their exact difference then?

We apologise for the lack of clarity in our expression here. The reviewer is right that both populations are proliferative. EpS1 represents a highly proliferative population of cells enriched in expression cell cycle markers, with GO terms relating to DNA replication, centromere complex assembly and chromosome organisation. By contrast, EpS2 represents cells are enriched in markers of the proliferative phase endometrium with GO terms including ‘reproductive structure development’, ‘epithelial cell development’ and ‘response to hormone’.

What are the in vivo counterparts of both subclusters? Same question applies to the stromal subclusters SS1 and SS2 (lines 206-207).

Recently we reported the characterisation of a population of highly proliferative mesenchymal cells (hPMC) in mid-luteal endometrial samples (PMID: 33764639). As our assembloids were established from luteal-phase biopsies, the SS1 population seen in our present data may be the in vitro representation of this mesenchymal population, i.e. clonogenic cells which retain a partially conserved signature of bone marrow-derived decidual progenitors. The SS2 population, by contrast, represents the resident stromal cells of the proliferative phase endometrium. Markers for endometrial epithelial progenitors are less well described. We hypothesise that the cells in EpS1 may represent a clonogenic population of epithelial cells, which have been shown to give rise to organoids when cultured as single cells (PMID: 28394884, PMID: 28442471) but this remains to be confirmed. Whether this population differs in patients with reproductive dysfunction (e.g. recurrent miscarriages) is also of interest to future studies.

– Line 180: canonical endometrial receptivity genes: please indicate them in the figure, or mention a few here in the text.

In both panels C and E of Figure 3, these genes are highlighted in green font, as indicated in the legend. We have now added the citations to the legend as well, per the request below.

– Lines 193 …: regarding the transitional population (TP): what may be the in vivo correlate? What could be the in vivo function of the TP? The TP appears not to be found in the in vivo endometrium (see Figure 3C). Line 257: there is a lack of crosstalk of TP with other cells, so is the TP an in vitro population not present in vivo? Should be discussed.

Based on single-cell analysis of fresh biopsies (i.e. processed immediately after collection), we reported the presence of an analogous, ambiguous population of cells co-expressing epithelial and stromal marker genes in midluteal endometrium (PMID: 29227245). As outlined in our response to similar questions raised by Reviewer 2, it is plausible that this transitional population engages in turnover of luminal epithelium during implantation and there is robust evidence that MET drives re-epithelisation following the onset of menstruation (see additional comments below).

– Line 218: "… novel pre-decidual genes were also identified" … moderate to "novel candidate pre-decidual genes …".

We have amended the statement, but like to note that both *DDIT4* (PMID: 29447340) and *P4HA2* (PMID: 25781565) are upregulated in vitro prior to the emergence of mature decidual cell markers, and in the case of *DDIT4* is critical for decidualization. Our exploration of publicly available microarray data (GEO dataset GDS2052 and manuscript Figure 3C) confirms their expression in early luteal phase endometrium, prior to decidualisation. Examination of these markers in the context of decidualization is otherwise lacking, but they appear to be exciting future targets.

– Lines 247…: "… transition between cellular states is predicated …". This could be further supported by applying pseudotime analysis (using Monocle, RNAvelocity, …). Did the authors consider or perform this?

Thank you, we did consider this suggestion. We made several attempts at pseudotime analysis of our dataset using different approaches but found that it did not perform well in the presence of only two timepoints (Day 0 and Day 4). We do consistently observe in such approaches that the (progesterone-resistant) senescent cells link closely with undifferentiated populations, concordant with shared pathways functioning to prepare for tissue renewal after menstruation.

– Line 299: dasatinib needs somewhat more introduction/explanation here. How does it work? What does it exactly do?

Thank you, we have amended the text to introduce the main target pathways of dasatinib.

– Line 315: TP develop by MET: what are the predicted source cells then? No relevant information available from pseudotime analysis?

As mentioned above, pseudotime analysis was not informative when applied to this dataset. However, based on the reduction in TP numbers and corresponding increase in decidual cells after dasatinib treatment, we propose that these cells arise after divergence of pre-decidual cells (SS3) at differentiation. The apparent dependence of TP cells on tyrosine kinase signalling agrees with developmental models of mesenchymal-to-epithelial transition (PMID: 13678588). Mesenchymal-to-epithelial transition is a well-described phenomenon in tissue differentiation and repair processes, and fundamental to endometrial biology (see PMID: 30407544, PMID: 23216285). The use of single-cell analysis, combined with our dasatinib experiments, enabled us to serendipitously identify this population.

– Line 328-331: this conclusion is not clear from the data; please rephrase and/or include in graphical abstract.

A graphical abstract is now included.

– Line 342: please immediately specify that exactly one embryo is added per assembloid drop/well.

Thank you, we have amended the text.

– Line 358: all embryo cocultures were found to secrete hCG; thus the 'receptive' co-cultures (without dasatinib) do not show more features of correct interaction ('pregnancy') than the dasatinib-treated ones? Please discuss.

The comparable levels of hCG secretion suggest equivalent blastocyst quality across the co-cultures, supporting our assertion that imbalance in decidual subpopulations is responsible for entrapment and collapse of embryos in dasatinib-treated assembloids (manuscript Figure 6D). Note, however, that the media was not changed during embryo co-culture, so we do not have temporal data on hCG expression for these embryos and the data presented are merely snapshots.

– Some typo's: line 259: EpS5; line 276: EpS4.

Thank you, we have corrected these errors.

Discussion:– Lines 364-373: This part is not really to the point, and could be removed to better retain the focus.

The point we intended to convey is that estrogen-dependent proliferation of epithelial and stromal cells is spatially controlled in the endometrium, leading to accumulation of replicative damaged cells underneath the luminal epithelium, i.e. the site of embryo implantation and tissue remodelling. However, we have amended the Discussion as requested.

– Line 423: "… MET drives re-epithelialization …". how sure is this in the field? Has this recently been unequivocally underpinned? If not, please moderate.

At menstruation, concurrent piecemeal repair of the luminal epithelium takes place adjacent to shedding functionalis (PMID: 19252193). Emerging cells exhibit features of migratory capacity characteristic to fibroblasts (i.e. intracellular microtubular systems and pseudopodial projections), but basement membrane formation and intercellular desmosomes in line with epithelial function (PMID: 2064209). These luminal EpC closely relate to the underlying EnSC, therefore mesenchymal to epithelial transition (MET) is the most likely mechanism to explain the rapid repair process (PMID: 30407544) since tissue shedding largely precludes the likelihood that other LE progenitors are present.

Methods:– Line 526-527: if I understand well, authors use 95% Matrigel/5% cells in medium (1:20), which is high and completely different from other studies using lower % of Matrigel; please confirm or explain.

Our protocol uses a Cells-to-Matrigel ratio of 1:20 (v/v) as described by Turco and colleagues (PMID: 28394884) in their method for the establishment of gland-like organoids.

– ELISAs: how were secretions in the medium normalized to the number of organoids/cells present in the assembloid drop? Were they normalized to extracted cellular protein b-actin? Or in other words, how was the number of organoids/cells in the structures standardized among wells/plates/independent experiments for secreted protein measurements?

We controlled the seeding density of assembloid cultures as described above and in the manuscript. Cells were harvested entirely for staining or single-cell sequencing experiments, so we were unable to extract cellular protein from these cohorts. ELISAs were performed in secretions from samples which were grown in parallel (i.e. not sequential) experiments so densities could be assessed objectively. Statistical analysis took pairing into account.

Figures:– Figure 2: PAEP and SPP1 in ExM/E2/cAMP/MPA are indicated as non-significantly different; is this true or correct?

We agree that the lack of statistical significance was unexpected. We have reviewed the statistical analysis (Friedman test and Dunn’s multiple comparisons *post-hoc* test, significant accepted at *P*<0.05) for these data and the results presented are correct. This is likely due to the spread of the data (note the logarithmic axis), despite consistent induction of *SPP1* in all 3 cultures. We used primary cells in these experiments, thus patient variability is the probable source of variance – however induction was seen across all patients, independent of phenotypic differences, indicating a robust response within the model. Culture in the NAC+ media was the only condition which appeared to support induction of both markers (regardless of the statistical outcome) and thus was selected for further experiments.

– Figure 3 C and E: in vivo: please add reference of these data in the legend.

Thank you, now added.

– Figure 6D: right figure is not explained.

Figure 6D shows the changes in diameters of embryos (µm) over 72 hours of time-lapse imaging, as determined in ImageJ. We have amended the legend to clarify this.

– Figure 6: "… attached by proliferating polar trophectoderm": is "proliferating" supported by experimental evidence? Please show then.

The polar trophectoderm of this embryo became multi-layered, hence our interpretation that proliferation is/has taken place. We agree that this requires further confirmation and will be accounted for in future immunolabelling experiments.

Reviewer #4 (Recommendations for the authors):Suggestions for improved or additional experiments– The authors should consider testing their minimal differentiation medium to determine whether it's sufficient to induce a stromal cell response, rather than only testing the secretory response of the endometrial epithelium. If these tests have been done, the authors should consider mentioning them at the place in the text where the epithelial secretion response is discussed.

See response above (Essential Revisions, point 3).

– Line 166 and 167 mentions the dotted line in figure 3B. Please add an explanation for the dotted line circles.

Thank you. The dotted circles were intended to highlight ciliated and TP populations. These populations were not segregated by the UMAP co-ordinates referenced, which we note broadly segregated the populations into stroma/epithelium and Day0/Day4. We have amended the Figure legend to make this clear.

– Line 172-173: Please provide more clarity on the difference between actively dividing epithelial cells and epithelial cells expressing markers of proliferating endometrium.

We apologise for the lack of clarity. EpS1 represents a highly proliferative epithelial population, enriched in expression cell cycle genes and functional GO terms relating to DNA replication, centromere complex assembly and chromosome organisation. By contrast, EpS2 represents cells enriched for proliferative phase endometrial genes with GO terms including ‘reproductive structure development’, ‘epithelial cell development’ and ‘response to hormone’.

– Line 218: Were these novel pre-decidual genes verified in human pre-decidual samples? Can the authors confirm that they are not artifacts of their culture system?

We thank the Reviewer for this query. Actually, upon closer scrutiny of the literature, both *DDIT4* (PMID: 29447340) and *P4HA2* (PMID: 25781565) have already been implicated in the decidualization process, although not as yet mapped to pre-decidual cells. Mining of publicly available microarray data (GEO dataset GDS2052 and manuscript Figure 3E) confirmed their expression in early luteal phase endometrium.

– Please clarify how ligands and receptors were chosen to be entered into the CellPhoneDB program.

CellPhoneDB is an online, publicly available repository of ligands, receptors and their interactions (https://github.com/Teichlab/cellphonedb), which integrates various existing datasets and new manually reviewed information. In order to use this computational tool, expression counts and cell metadata were extracted from our single-cell data for decidualizing cells (i.e. those populations present in D4 cultures) according to pipelines provided by the CellPhoneDB vignette. The CellphoneDB package then derives enriched receptor–ligand interactions between two cell types based on expression of a receptor by one cell type and a ligand by another cell type (as described here: https://www.cellphonedb.org/explore-sc-rna-seq). We chose to exclude integrin-interactions from our analysis and focus on cell-cell interactions rather than cell-ECM, but future investigation of these is certainly of interest to the progression of the model.

– Line 260: Please clarify the CellPhoneDB data. It appears that only the RNA expression data is used for this experiment, so can it be assumed that the two proteins are both interacting, even if they are both expressed? The CellPhoneDB data should be validated.

CellphoneDB is a curated repository of ligands and receptors, and their interactions, which is used to predict biologically relevant interacting ligand-receptor partners computationally from single-cell transcriptomics (scRNAseq) data. During the analysis pipeline, CellphoneDB considers the absolute expression levels of ligands and receptors within each cell type (using counts data) and applies empirical shuffling to calculate which ligand–receptor pairs display significant cell-type specificity. This predicts molecular interactions between cell populations via specific protein complexes and generates potential cell–cell communication networks. The dasatinib experiments were based on CellphoneDB predictions and in, that sense, constitute a validation experiment.

– Line 316: Why does the fact that dasatinib upregulates mesenchymal genes and decidual factors in transitional cells show that MET causes the transitional cells to emerge during decidualization? More evidence is needed for this claim.

We agree with the Reviewer. We amended the text and now state that the data only suggest that these cells are of stromal origins. This conjecture is supported indirectly by studies highlighted the critical role of MET in endometrial repair.

– Line 361: Please clarify which different pathological endometrial states were recapitulated.

Thank you, we have elaborated this statement.

– Is there data demonstrating that implantation failure is associated with insufficient senescence?

Yes, direct evidence has come from Cornelis Lambalk and colleagues who demonstrated in a prospective cohort study premature expression of the decidualization marker PRL is associated with repeated implantation failure (RIF) (PMID: 31389284). This study was based on immunohistochemistry. At mRNA level, PRL is not a particularly useful marker to analyse the decidual response in whole endometrial samples. We recently reported much more specific marker genes selectively expressed in decidualising stromal cells (e.g. *SCARA5*) and senescent decidual cells (e.g. *DIO2*) in luteal endometrium. We can share with the Reviewer that an ongoing analysis of almost 800 endometrial biopsies indicates that RIF is associated with increased frequency of cycles characterised by excessive decidualisation (high SCARA5, low DIO2). Conversely, RPL is associated with excessive senescence (high DIO2, low SCARA5) and the frequency of cycles with excessive senescence increases stepwise with the number of previous miscarriages, and thus the recurrence risk of miscarriage.

– The "assembloids" are new but there are renditions of the 3D models using epithelial cells and stromal cells that are published and could be cited for comparison purposes. Given the different way the stromal cells were cocultured (in hydrogel) in the assembloid, the morphology is unique and the transcriptomics are new.

Thank you. We are pleased to inform the Reviewer that we now have a comprehensive review of endometrial co-culture models under review.

– Single cell RNA-seq of organoids have been performed by Spencer's group although they did not have the cocultured stromal cells. In that sense, the scRNA-seq of epithelial/stromal assembloids have revealed new data and it would be interesting to determine how epithelial transcriptomics changed in the presence of stromal cells.

We agree that comparison between the transcriptomes of epithelial organoid cultures and epithelial/stromal assembloids would be of interest both to ourselves and others. Unfortunately, the single-cell data for the previously published results are not publicly available. Differences in the culture conditions (matrix, media etc) may also confound comparisons between datasets generated in different labs. As mentioned in our response to Reviewer 1, we are in the process of assessing if stromal cells from control subjects can attenuate epithelial defects in gland-like organoids of women with recurrent missed miscarriage.

[Editors' note: further revisions were suggested prior to acceptance, as described below.]

The reviewers have discussed their reviews with one another, and the Reviewing Editor has drafted this to help you prepare a revised submission.Essential revisions:Reviewer #4 (Recommendations for the authors):The authors have addressed most of Reviewer 4's concerns. However, it remains important that other epithelial and stromal organoid systems are cited in this study, as also suggested by Reviewer 3.

Other key papers are now cited in both the introduction and discussion.

Overall, this is a lovely study.

Thank you very much.